# Reinforcement Learning under Model Mismatch

Aurko Roy[1], Huan Xu[2], and Sebastian Pokutta[2]

[1]Google,[*] *Email:* `aurkor@google.com`
[2]ISyE, Georgia Institute of Technology, Atlanta, GA, USA.
*Email:* `huan.xu@isye.gatech.edu`
[2]ISyE, Georgia Institute of Technology, Atlanta, GA, USA.
*Email:* `sebastian.pokutta@isye.gatech.edu`

## Abstract

We study reinforcement learning under *model misspecification*, where we do not have access to the true environment but only to a reasonably close approximation to it. We address this problem by extending the framework of robust MDPs of [1, 15, 11] to the *model-free* Reinforcement Learning setting, where we do not have access to the model parameters, but can only sample states from it. We define *robust versions* of Q-learning, SARSA, and TD-learning and prove convergence to an approximately optimal robust policy and approximate value function respectively. We scale up the robust algorithms to large MDPs via function approximation and prove convergence under two different settings. We prove convergence of robust approximate policy iteration and robust approximate value iteration for linear architectures (under mild assumptions). We also define a robust loss function, the *mean squared robust projected Bellman error* and give stochastic gradient descent algorithms that are guaranteed to converge to a local minimum.

## 1 Introduction

Reinforcement learning is concerned with learning a good policy for sequential decision making problems modeled as a Markov Decision Process (MDP), via interacting with the environment [20, 18]. In this work we address the problem of reinforcement learning from a *misspecified model*. As a motivating example, consider the scenario where the problem of interest is not directly accessible, but instead the agent can interact with a simulator whose dynamics is reasonably close to the true problem. Another plausible application is when the parameters of the model may evolve over time but can still be reasonably approximated by an MDP.

To address this problem we use the framework of *robust MDPs* which was proposed by [1, 15, 11] to solve the planning problem under model misspecification. The robust MDP framework considers a class of models and finds the robust optimal policy which is a policy that performs best under the worst model. It was shown by [1, 15, 11] that the robust optimal policy satisfies the *robust Bellman equation* which naturally leads to exact dynamic programming algorithms to find an optimal policy. However, this approach is model dependent and does not immediately generalize to the model-free case where the parameters of the model are unknown.

Essentially, reinforcement learning is a *model-free* framework to solve the Bellman equation using samples. Therefore, to learn policies from misspecified models, we develop sample based methods to solve the *robust* Bellman equation. In particular, we develop robust versions of classical reinforcement learning algorithms such as Q-learning, SARSA, and TD-learning and prove convergence to an approximately optimal policy under mild assumptions on the discount factor. We also show that

---

[*]Work done while at Georgia Tech

the nominal versions of these iterative algorithms converge to policies that may be arbitrarily worse compared to the optimal policy.

We also scale up these robust algorithms to large scale MDPs via function approximation, where we prove convergence under two different settings. Under a technical assumption similar to [5, 24] we show convergence of robust approximate policy iteration and value iteration algorithms for linear architectures. We also study function approximation with nonlinear architectures, by defining an appropriate *mean squared robust projected Bellman error* (MSRPBE) loss function, which is a generalization of the mean squared projected Bellman error (MSPBE) loss function of [22, 21, 6]. We propose robust versions of stochastic gradient descent algorithms as in [22, 21, 6] and prove convergence to a local minimum under some assumptions for function approximation with arbitrary smooth functions.

**Contribution.**   In summary we have the following contributions:

1. We extend the robust MDP framework of [1, 15, 11] to the *model-free* reinforcement learning setting. We then define robust versions of Q-learning, SARSA, and TD-learning and prove convergence to an approximately optimal robust policy.

2. We also provide robust reinforcement learning algorithms for the function approximation case and prove convergence of robust approximate policy iteration and value iteration algorithms for linear architectures. We also define the MSRPBE loss function which contains the robust optimal policy as a local minimum and we derive stochastic gradient descent algorithms to minimize this loss function as well as establish convergence to a local minimum in the case of function approximation by arbitrary smooth functions.

3. Finally, we demonstrate empirically the improvement in performance for the robust algorithms compared to their nominal counterparts. For this we used various Reinforcement Learning test environments from OpenAI [9] as benchmark to assess the improvement in performance as well as to ensure reproducibility and consistency of our results.

**Related Work.**   Recently, several approaches have been proposed to address model performance due to parameter uncertainty for Markov Decision Processes (MDPs). A Bayesian approach was proposed by [19] which requires perfect knowledge of the prior distribution on transition matrices. Other probabilistic and risk based settings were studied by [10, 25, 23] which propose various mechanisms to incorporate percentile risk into the model. A framework for robust MDPs was first proposed by [1, 15, 11] who consider the transition matrices to lie in some *uncertainty set* and proposed a dynamic programming algorithm to solve the robust MDP. Recent work by [24] extended the robust MDP framework to the function approximation setting where under a technical assumption the authors prove convergence to an optimal policy for linear architectures. Note that these algorithms for robust MDPs do not readily generalize to the *model-free* reinforcement learning setting where the parameters of the environment are not explicitly known.

For reinforcement learning in the non-robust *model-free* setting, several iterative algorithms such as Q-learning, TD-learning, and SARSA are known to converge to an optimal policy under mild assumptions, see [4] for a survey. Robustness in reinforcement learning for MDPs was studied by [13] who introduced a robust learning framework for learning with disturbances. Similarly, [16] also studied learning in the presence of an adversary who might apply disturbances to the system. However, for the algorithms proposed in [13, 16] no theoretical guarantees are known and there is only limited empirical evidence. Another recent work on robust reinforcement learning is [12], where the authors propose an online algorithm with certain transitions being stochastic and the others being adversarial and the devised algorithm ensures low regret.

For the case of reinforcement learning with large MDPs using function approximations, theoretical guarantees for most TD-learning based algorithms are only known for linear architectures [2]. Recent work by [6] extended the results of [22, 21] and proved that a stochastic gradient descent algorithm minimizing the *mean squared projected Bellman equation* (MSPBE) loss function converges to a local minimum, even for nonlinear architectures. However, these algorithms do not apply to robust MDPs; in this work we extend these algorithms to the robust setting.

## 2 Preliminaries

We consider an infinite horizon Markov Decision Process (MDP) [18] with finite state space $\mathcal{X}$ of size $n$ and finite action space $\mathcal{A}$ of size $m$. At every time step $t$ the agent is in a state $i \in \mathcal{X}$ and can choose an action $a \in \mathcal{A}$ incurring a cost $c_t(i,a)$. We will make the standard assumption that future cost is discounted, see e.g., [20], with a discount factor $\vartheta < 1$ applied to future costs, i.e., $c_t(i,a) := \vartheta^t c(i,a)$, where $c(i,a)$ is a fixed constant independent of the time step $t$ for $i \in \mathcal{X}$ and $a \in \mathcal{A}$. The states transition according to probability transition matrices $\tau := \{P^a\}_{a \in \mathcal{A}}$ which depends only on their last taken action $a$. A *policy of the agent* is a sequence $\pi = (\mathbf{a_0}, \mathbf{a_1}, \dots)$, where every $\mathbf{a_t}(i)$ corresponds to an action in $\mathcal{A}$ if the system is in state $i$ at time $t$. For every policy $\pi$, we have a corresponding value function $v_\pi \in \mathbb{R}^n$, where $v_\pi(i)$ for a state $i \in \mathcal{X}$ measures the expected cost of that state if the agent were to follow policy $\pi$. This can be expressed by the recurrence relation

$$v_\pi(i) := c(i, \mathbf{a_0}(i)) + \vartheta \mathbb{E}_{j \sim \mathcal{X}} [v_\pi(j)].\tag{1}$$

The goal is to devise algorithms to learn an optimal policy $\pi^*$ that minimizes the expected total cost:

**Definition 2.1** (Optimal policy). *Given an MDP with state space $\mathcal{X}$, action space $\mathcal{A}$ and transition matrices $P^a$, let $\Pi$ be the strategy space of all possibile policies. Then an optimal policy $\pi^*$ is one that minimizes the expected total cost, i.e.,*

$$\pi^* := \arg\min_{\pi \in \Pi} \mathbb{E} \left[ \sum_{t=0}^{\infty} \vartheta^t c(i_t, \mathbf{a_t}(i_t)) \right].\tag{2}$$

In the robust case we will assume as in [15, 11] that the transition matrices $P^a$ are not fixed and may come from some uncertainty region $\mathcal{P}^a$ and may be chosen adversarially by nature in future runs of the model. In this setting, [15, 11] prove the following *robust* analogue of the *Bellman recursion*. A *policy of nature* is a sequence $\tau := (\mathbf{P_0}, \mathbf{P_1}, \dots)$ where every $P_t(a) \in \mathcal{P}^a$ corresponds to a transition probability matrix chosen from $\mathcal{P}^a$. Let $\mathcal{T}$ denote the set of all such policies of nature. In other words, a policy $\tau \in \mathcal{T}$ of nature is a sequence of transition matrices that may be played by it in response to the actions of the agent. For any set $P \subseteq \mathbb{R}^n$ and vector $v \in \mathbb{R}^n$, let $\sigma_P(v) := \sup \{p^\top v \mid p \in P\}$ be the *support function* of the set $P$. For a state $i \in \mathcal{X}$, let $\mathcal{P}_i^a$ be the projection onto the $i^{th}$ row of $\mathcal{P}^a$.

**Theorem 2.2.** *[15] We have the following perfect duality relation*

$$\min_{\pi \in \Pi} \max_{\tau \in \mathcal{T}} \mathbb{E}_\tau \left[ \sum_{t=0}^{\infty} \vartheta^t c(i_t, \mathbf{a_t}(i_t)) \right] = \max_{\tau \in \mathcal{T}} \min_{\pi \in \Pi} \mathbb{E}_\tau \left[ \sum_{t=0}^{\infty} \vartheta^t c(i_t, \mathbf{a_t}(i_t)) \right].\tag{3}$$

*The optimal value function $v_{\pi^*}$ corresponding to the optimal policy $\pi^*$ satisfies*

$$v_{\pi^*}(i) = \min_{a \in \mathcal{A}} \left( c(i,a) + \vartheta \sigma_{\mathcal{P}_i^a}(v_{\pi^*}) \right),\tag{4}$$

*and $\pi^*$ can then be obtained in a greedy fashion, i.e.,*

$$\mathbf{a}^*(i) \in \arg\min_{a \in \mathcal{A}} \left\{ c(i,a) + \vartheta \sigma_{\mathcal{P}_i^a}(v) \right\}.\tag{5}$$

The main shortcoming of this approach is that it does not generalize to the *model free* case where the transition probabilities are not explicitly known but rather the agent can only sample states according to these probabilities. In the absence of this knowledge, we cannot compute the support functions of the uncertainty sets $\mathcal{P}_i^a$. On the other hand it is often easy to have a *confidence region* $U_i^a$, e.g., a ball or an ellipsoid, corresponding to every state-action pair $i \in \mathcal{X}, a \in \mathcal{A}$ that quantifies our uncertainty in the simulation, with the uncertainty set $\mathcal{P}_i^a$ being the confidence region $U_i^a$ centered around the unknown simulator probabilities. Formally, we define the uncertainty sets corresponding to every state action pair in the following fashion.

**Definition 2.3** (Uncertainty sets). *Corresponding to every state-action pair $(i,a)$ we have a* confidence *region $U_i^a$ so that the uncertainty region $\mathcal{P}_i^a$ of the probability transition matrix corresponding to $(i,a)$ is defined as*

$$\mathcal{P}_i^a := \{x + p_i^a \mid x \in U_i^a\},\tag{6}$$

*where $p_i^a$ is the* unknown *state transition probability vector from the state $i \in \mathcal{X}$ to every other state in $\mathcal{X}$ given action $a$ during the simulation.*

As a simple example, we have the ellipsoid $U_i^a := \{x \mid x^\top A_i^a x \leq 1, \sum_{i \in \mathcal{X}} x_i = 0\}$ for some $n \times n$ psd matrix $A_i^a$ with the uncertainty set $\mathcal{P}_i^a$ being $\mathcal{P}_i^a := \{x + p_i^a \mid x \in U_i^a\}$, where $p_i^a$ is the *unknown* simulator state transition probability vector with which the agent transitioned to a new state during training. Note that while it may easy to come up with good descriptions of the confidence region $U_i^a$, the approach of [15, 11] breaks down since we have no knowledge of $p_i^a$ and merely observe the new state $j$ sampled from this distribution.

In the following sections we develop *robust versions* of Q-learning, SARSA, and TD-learning which are guaranteed to converge to an approximately optimal policy that is robust with respect to this confidence region. The robust versions of these iterative algorithms involve an additional linear optimization step over the set $U_i^a$, which in the case of $U_i^a = \{\|x\|_2 \leq r\}$ simply corresponds to adding fixed noise during every update. In later sections we will extend it to the function approximation case where we study linear architectures as well as nonlinear architectures; in the latter case we derive new stochastic gradient descent algorithms for computing approximately robust policies.

## 3 Robust exact dynamic programming algorithms

In this section we develop robust versions of exact dynamic programming algorithms such as Q-learning, SARSA, and TD-learning. These methods are suitable for small MDPs where the size $n$ of the state space is not too large. Note that confidence region $U_i^a$ must also be constrained to lie within the probability simplex $\Delta_n$. However since we do not have knowledge of the simulator probabilities $p_i^a$, we do not know how far away $p_i^a$ is from the boundary of $\Delta_n$ and so the algorithms will make use of a proxy confidence region $\widehat{U_i^a}$ where we drop the requirement of $\widehat{U_i^a} \subseteq \Delta_n$, to compute the robust optimal policies. With a suitable choice of step lengths and discount factors we can prove convergence to an approximately optimal $U_i^a$-robust policy where the approximation depends on the difference between the unconstrained proxy region $\widehat{U_i^a}$ and the true confidence region $U_i^a$. Below we give specific examples of possible choices for simple confidence regions.

**Ellipsoid:** Let $\{A_i^a\}_{i,a}$ be a sequence of $n \times n$ psd matrices. Then we can define the confidence region as

$$U_i^a := \left\{ x \,\middle|\, x^\top A_i^a x \leq 1, \sum_{i \in \mathcal{X}} x_i = 0, -p_{ij}^a \leq x_j \leq 1 - p_{ij}^a, \forall j \in \mathcal{X} \right\}. \tag{7}$$

Note that $U_i^a$ has some additional linear constraints so that the uncertainty set $\mathcal{P}_i^a := \{p_i^a + x \mid x \in U_i^a\}$ lies inside $\Delta_n$. Since we do not know $p_i^a$, we will make use of the proxy confidence region $\widehat{U_i^a} := \{x \mid x^\top A_i^a x \leq 1, \sum_{i \in \mathcal{X}} x_i = 0\}$. In particular when $A_i^a = r^{-1} I_n$ for every $i \in \mathcal{X}, a \in \mathcal{A}$ then this corresponds to a spherical confidence interval of $[-r, r]$ in every direction. In other words, each uncertainty set $\mathcal{P}_i^a$ is an $\ell_2$ ball of radius $r$.

**Parallelepiped:** Let $\{B_i^a\}_{i,a}$ be a sequence of $n \times n$ invertible matrices. Then we can define the confidence region as

$$U_i^a := \left\{ x \,\middle|\, \|B_i^a x\|_1 \leq 1, \sum_{i \in \mathcal{X}} x_i = 0, -p_{ij}^a \leq x_j \leq 1 - p_{ij}^a, \forall j \in \mathcal{X} \right\}. \tag{8}$$

As before, we will use the unconstrained parallelepiped $\widehat{U_i^a}$ without the $-p_{ij}^a \leq x_j \leq 1 - p_{ij}^a$ constraints, as a proxy for $U_i^a$ since we do not have knowledge $p_i^a$. In particular if $B_i^a = D$ for a diagonal matrix $D$, then the proxy confidence region $\widehat{U_i^a}$ corresponds to a rectangle. In particular if every diagonal entry is $r$, then every uncertainty set $\mathcal{P}_i^a$ is an $\ell_1$ ball of radius $r$.

### 3.1 Robust Q-learning

Let us recall the notion of a Q-factor of a state-action pair $(i, a)$ and a policy $\pi$ which in the non-robust setting is defined as

$$Q(i, a) := c(i, a) + \mathbb{E}_{j \sim \mathcal{X}} [v(j)], \tag{9}$$

where $v$ is the value function of the policy $\pi$. In other words, the Q-factor represents the expected cost if we start at state $i$, use the action $a$ and follow the policy $\pi$ subsequently. One may similarly define the *robust* Q-factors using a similar interpretation and the minimax characterization of Theorem 2.2. Let $Q^*$ denote the Q-factors of the optimal robust policy and let $v^* \in \mathbb{R}^n$ be its value function. Note that we may write the value function in terms of the Q-factors as $v^* = \min_{a \in \mathcal{A}} Q^*(i, a)$. From Theorem 2.2 we have the following expression for $Q^*$:

$$Q^*(i, a) = c(i, a) + \vartheta \sigma_{\mathcal{P}_i^a}(v^*) \tag{10}$$

$$= c(i, a) + \vartheta \sigma_{U_i^a}(v^*) + \vartheta \sum_{j \in \mathcal{X}} p_{ij}^a \min_{a' \in \mathcal{A}} Q^*(j, a'), \tag{11}$$

where equation (11) follows from Definition 2.3. For an estimate $Q_t$ of $Q^*$, let $v_t \in \mathbb{R}^n$ be its value vector, i.e., $v_t(i) := \min_{a \in \mathcal{A}} Q_t(i, a)$. The *robust* Q-*iteration* is defined as:

$$Q_t(i, a) := (1 - \gamma_t) Q_{t-1}(i, a) + \gamma_t \left( c(i, a) + \vartheta \sigma_{\widehat{U_i^a}}(v_{t-1}) + \vartheta \min_{a' \in \mathcal{A}} Q_{t-1}(j, a') \right), \tag{12}$$

where a state $j \in \mathcal{X}$ is sampled with the unknown transition probability $p_{ij}^a$ using the simulator. Note that the robust Q-iteration of equation (12) involves an additional linear optimization step to compute the support function $\sigma_{\widehat{U_i^a}}(v_t)$ of $v_t$ over the proxy confidence region $\widehat{U_i^a}$. We will prove that iterating equation (12) converges to an approximately optimal policy. The following definition introduces the notion of an $\varepsilon$-optimal policy, see e.g., [4]. The error factor $\varepsilon$ is also referred to as the *amplification factor*. We will treat the Q-factors as a $|\mathcal{X}| \times |\mathcal{A}|$ matrix in the definition so that its $\ell_\infty$ norm is defined as usual.

**Definition 3.1** ($\varepsilon$-optimal policy). *A policy $\pi$ with Q-factors $Q'$ is $\varepsilon$-optimal with respect to the optimal policy $\pi^*$ with corresponding Q-factors $Q^*$ if $\left\| Q' - Q^* \right\|_\infty \leq \varepsilon \left\| Q^* \right\|_\infty$.*

The following simple lemma allows us to decompose the optimization of a linear function over the proxy uncertainty set $\widehat{\mathcal{P}_i^a}$ in terms of linear optimization over $\mathcal{P}_i^a$, $U_i^a$, and $\widehat{U_i^a}$.

**Lemma 3.2.** *Let $v \in \mathbb{R}^n$ be any vector and let $\beta_i^a := \max_{y \in \widehat{U_i^a}} \min_{x \in U_i^a} \|y - x\|_1$. Then we have*

$$\sigma_{\widehat{\mathcal{P}_i^a}}(v) \leq \sigma_{\mathcal{P}_i^a}(v) + \beta_i^a \|v\|_\infty.$$

The following theorem proves that under a suitable choice of step lengths $\gamma_t$ and discount factor $\vartheta$, the iteration of equation (12) converges to an $\varepsilon$-approximately optimal policy with respect to the confidence regions $U_i^a$.

**Theorem 3.3.** *Let the step lengths $\gamma_t$ of the Q-iteration algorithm be chosen such that $\sum_{t=0}^{\infty} \gamma_t = \infty$ and $\sum_{t=0}^{\infty} \gamma_t^2 < \infty$ and let the discount factor $\vartheta < 1$. Let $\beta_i^a$ be as in Lemma 3.2 and let $\beta := \max_{i \in \mathcal{X}, a \in \mathcal{A}} \beta_i^a$. If $\vartheta(1 + \beta) < 1$ then with probability 1 the iteration of equation (12) converges to an $\varepsilon$-optimal policy where $\varepsilon := \frac{\vartheta \beta}{1 - \vartheta(1 + \beta)}$.*

**Remark 3.4.** *If $\beta = 0$ then note that by Theorem 3.3, the robust Q-iterations converge to the exact optimal Q-factors since $\varepsilon = 0$. Since $\beta_i^a := \max_{y \in \widehat{U_i^a}} \min_{x \in U_i^a} \|y - x\|_1$, it follows that $\beta = 0$ iff $\widehat{U_i^a} = U_i^a$ for every $i \in \mathcal{X}, a \in \mathcal{A}$. This happens when the confidence region is small enough so that the simplex constraints $-p_{ij}^a \leq x_j \leq 1 - p_{ij}^a \forall j \in \mathcal{X}$ in the description of $\mathcal{P}_i^a$ become redundant for every $i \in \mathcal{X}, a \in \mathcal{A}$. Equivalently every $p_i^a$ is "far" from the boundary of the simplex $\Delta_n$ compared to the size of the confidence region $U_i^a$.*

**Remark 3.5.** *Note that simply using the nominal Q-iteration without the $\sigma_{\widehat{U_i^a}}(v)$ term does not guarantee convergence to $Q^*$. Indeed, the nominal Q-iterations converge to Q-factors $Q'$ where $\left\| Q' - Q^* \right\|_\infty$ may be arbitrary large. This follows easily from observing that*

$$\left| Q'(i, a) - Q^*(i, a) \right| = \left| \sigma_{\widehat{U_i^a}}(v^*) \right| \tag{13}$$

*, where $v^*$ is the value function of $Q^*$ and so*

$$\left\| Q' - Q^* \right\|_\infty = \max_{i \in \mathcal{X}, a \in \mathcal{A}} \left| \sigma_{\widehat{U_i^a}}(v^*) \right| \tag{14}$$

*which can be as high as $\|v^*\|_\infty = \|Q^*\|_\infty$.*

## 3.2 Robust TD-Learning

Let $(i_0, i_1, \dots)$ be a trajectory of the agent, where $i_m$ denotes the state of the agent at time step $m$. The main idea behind the TD($\lambda$)-learning method is to estimate the value function $v_\pi$ of a policy $\pi$ using the *temporal difference* errors $d_m$ defined as

$$d_m := c(i_m, \pi(i_m)) + v v_t(i_{m+1}) - v_t(i_m). \tag{15}$$

For a parameter $\lambda \in (0, 1)$, the TD-learning iteration is defined in terms of the temporal difference errors as

$$v_{t+1}(i_k) := v_t(i_k) + \gamma_t \left( \sum_{m=k}^{\infty} (\vartheta\lambda)^{m-k} d_m \right). \tag{16}$$

In the robust setting, we have a confidence region $U_i^a$ with proxy $\widehat{U_i^a}$ for every temporal difference error, which leads us to define the *robust temporal difference* errors as

$$\widetilde{d_m} := d_m + \vartheta \sigma_{\widehat{U_{i_m}^{\pi(im)}}}(v_t), \tag{17}$$

where $d_m$ is the non-robust temporal difference. The *robust* TD-update is the usual TD-update, with the *robust temporal difference* errors $\widetilde{d_m}$ replacing the usual temporal difference error $d_m$. We define an $\varepsilon$-suboptimal value function for a fixed policy $\pi$ similar to Definition 3.1.

**Definition 3.6** ($\varepsilon$-approximate value function). *Given a policy $\pi$, we say that a vector $v' \in \mathbb{R}^n$ is an $\varepsilon$-approximation of $v_\pi$ if $\|v' - v_\pi\|_\infty \leq \varepsilon \|v_\pi\|_\infty$.*

The following theorem guarantees convergence of the robust TD-iteration to an approximate value function for $\pi$. We refer the reader to the supplementary material for a proof.

**Theorem 3.7.** *Let $\beta_i^a$ be as in Lemma 3.2 and let $\beta := \max_{i \in \mathcal{X}, a \in \mathcal{A}} \beta_i^a$. Let $\rho := \frac{\vartheta\lambda}{1-\vartheta\lambda}$. If $\vartheta(1 + \rho\beta) < 1$ then the robust TD-iteration converges to an $\varepsilon$-approximate value function, where $\varepsilon := \frac{\vartheta\beta}{1-\vartheta(1+\rho\beta)}$. In particular if $\beta_i^a = \beta = 0$, i.e., the proxy confidence region $\widehat{U_i^a}$ is the same as the true confidence region $U_i^a$, then the convergence is exact, i.e., $\varepsilon = 0$.*

# 4 Robust Reinforcement Learning with function approximation

In Section 3 we derived robust versions of exact dynamic programming algorithms such as Q-learning, SARSA and TD-learning respectively. If the state space $\mathcal{X}$ of the MDP is large then it is prohibitive to maintain a lookup table entry for every state. A standard approach for large scale MDPs is to use the *approximate dynamic programming* (ADP) framework [17]. In this setting, the problem is parametrized by a smaller dimensional vector $\theta \in \mathbb{R}^d$ where $d \ll n = |\mathcal{X}|$.

The natural generalizations of Q-learning, SARSA, and TD-learning algorithms of Section 3 are via the *projected Bellman equation*, where we project back to the space spanned by all the parameters in $\theta \in \mathbb{R}^d$, since they are the value functions representable by the model. Convergence for these algorithms even in the non-robust setting are known only for linear architectures, see e.g., [2]. Recent work by [6] proposed stochastic gradient descent algorithms with convergence guarantees for smooth nonlinear function architectures, where the problem is framed in terms of minimizing a loss function. We give robust versions of both these approaches.

## 4.1 Robust approximations with linear architectures

In the approximate setting with linear architectures, we approximate the value function $v_\pi$ of a policy $\pi$ by $\Phi\theta$ where $\theta \in \mathbb{R}^d$ and $\Phi$ is a $n \times d$ *feature matrix* with rows $\phi(j)$ for every state $j \in \mathcal{X}$ representing its *feature vector*. Let $S$ be the span of the columns of $\Phi$, i.e., $S := \left\{ \Phi\theta \mid \theta \in \mathbb{R}^d \right\}$.

Define the operator $T_\pi : \mathbb{R}^n \to \mathbb{R}^n$ as $(T_\pi v)(i) := c(i, \pi(i)) + \vartheta \sum_{j \in \mathcal{X}} p_{ij}^{\pi(i)} v(j)$, so that the true value function $v_\pi$ satisfies $T_\pi v_\pi = v_\pi$. A natural approach towards estimating $v_\pi$ given a current estimate $\Phi\theta_t$ is to compute $T_\pi(\Phi\theta_t)$ and project it back to $S$ to get the next parameter $\theta_{t+1}$. The motivation behind such an iteration is the fact that the true value function is a fixed point of

this operation if it belonged to the subspace $S$. This gives rise to the *projected Bellman equation* where the projection $\Pi$ is typically taken with respect to a weighted Euclidean norm $\|\cdot\|_\xi$, i.e., $\|x\|_\xi = \sum_{i\in\mathcal{X}} \xi_i x_i^2$, where $\xi$ is some probability distribution over the states $\mathcal{X}$.

In the *model free* case, where we do not have explicit knowledge of the transition probabilities, various methods like LSTD($\lambda$), LSPE($\lambda$), TD($\lambda$) have been proposed [3, 8, 7, 14, 22, 21]. The key idea behind proving convergence for these methods is to show that the mapping $\Pi T_\pi$ is a contraction mapping with respect to the $\|\cdot\|_\xi$ for some distribution $\xi$ over the states $\mathcal{X}$. While the operator $T_\pi$ in the non-robust case is linear and is a contraction in the $\ell_\infty$ norm as in Section 3, the projection operator with respect to such norms is not guaranteed to be a contraction. However, it is known that if $\xi$ is the steady state distribution of the policy $\pi$ under evaluation, then $\Pi$ is non-expansive in $\|\cdot\|_\xi$ [4, 2].

In the robust setting, we have the same methods but with the *robust Bellman operators* $T_\pi$ defined as $(T_\pi v)(i) := c(i, \pi(i)) + \vartheta \sigma_{\mathcal{P}_i^{\pi(i)}}(v)$. Since we do not have access to the simulator probabilities $p_i^a$, we will use a proxy set $\widehat{\mathcal{P}_i^a}$ as in Section 3, with the proxy operator denoted by $\widehat{T_\pi}$. While the iterative methods of the non-robust setting generalize via the robust operator $T_\pi$ and the *robust projected Bellman equation* $\Phi\theta = \Pi T_\pi(\Phi\theta)$, it is however not clear how to choose the distribution $\xi$ under which the projected operator $\Pi T_\pi$ is a contraction in order to show convergence. Let $\xi$ be the steady state distribution of the *exploration policy* $\widehat{\pi}$ of the MDP with transition probability matrix $P^{\widehat{\pi}}$. We make the following assumption on the discount factor $\vartheta$ as in [24].

**Assumption 4.1.** *For every state $i \in \mathcal{X}$ and action $a \in \mathcal{A}$, there exists a constant $\alpha \in (0,1)$ such that for any $p \in \mathcal{P}_i^a$ we have $\vartheta p_j \leq \alpha P_{ij}^{\widehat{\pi}}$ for every $j \in \mathcal{X}$.*

Assumption 4.1 might appear artificially restrictive; however, it is necessary to prove that $\Pi T_\pi$ is a contraction. While [24] require this assumption for proving convergence of robust MDPs, a similar assumption is also required in proving convergence of *off-policy* Reinforcement Learning methods of [5] where the states are sampled from an exploration policy $\widehat{\pi}$ which is not necessarily the same as the policy $\pi$ under evaluation. Note that in the robust setting, all methods are necessarily *off-policy* since the transition matrices are not fixed for a given policy.

The following lemma is an $\xi$-weighted Euclidean norm version of Lemma 3.2.

**Lemma 4.2.** *Let $v \in \mathbb{R}^n$ be any vector and let $\beta_i^a := \dfrac{\max_{y\in\widehat{U_i^a}} \min_{x\in U_i^a} \|y-x\|_\xi}{\xi_{\min}}$. Then we have $\sigma_{\widehat{\mathcal{P}_i^a}}(v) \leq \sigma_{\mathcal{P}_i^a}(v) + \beta_i^a \|v\|_\xi$, where $\xi_{\min} := \min_{i\in\mathcal{X}} \xi_i$.*

The following theorem shows that the robust projected Bellman equation is a contraction under some assumptions on the discount factor $\vartheta$.

**Theorem 4.3.** *Let $\beta_i^a$ be as in Lemma 4.2 and let $\beta := \max_{i\in\mathcal{X}} \beta_i^{\pi(i)}$. If the discount factor $\vartheta$ satisfies Assumption 4.1 and $\alpha^2 + \vartheta^2\beta^2 < \frac{1}{2}$, then the operator $\widehat{T_\pi}$ is a contraction with respect to $\|\cdot\|_\xi$. In other words for any two $\theta, \theta' \in \mathbb{R}^d$, we have*

$$\left\| \widehat{T_\pi}(\Phi\theta) - \widehat{T_\pi}(\Phi\theta') \right\|_\xi^2 \leq 2\left(\alpha^2 + \vartheta^2\beta^2\right) \|\Phi\theta - \Phi\theta'\|_\xi^2 < \|\Phi\theta - \Phi\theta'\|_\xi^2. \qquad (18)$$

*If $\beta_i = \beta = 0$ so that $\widehat{U_i^{\pi(i)}} = U_i^{\pi(i)}$, then we have a simpler contraction under the assumption that $\alpha < 1$.*

The following corollary shows that the solution to the proxy projected Bellman equation converges to a solution that is not too far away from the true value function $v_\pi$.

**Corollary 4.4.** *Let Assumption 4.1 hold and let $\beta$ be as in Theorem 4.3. Let $\widetilde{v}_\pi$ be the fixed point of the projected Bellman equation for the proxy operator $\widehat{T_\pi}$, i.e., $\Pi\widehat{T_\pi}\widetilde{v}_\pi = \widetilde{v}_\pi$. Let $\widehat{v}_\pi$ be the fixed point of the proxy operator $\widehat{T_\pi}$, i.e., $\widehat{T_\pi}\widehat{v}_\pi = \widehat{v}_\pi$. Let $v_\pi$ be the true value function of the policy $\pi$, i.e., $T_\pi v_\pi = v_\pi$. Then it follows that*

$$\|\widetilde{v}_\pi - v_\pi\|_\xi \leq \frac{\vartheta\beta \|v_\pi\|_\xi + \|\Pi v_\pi - v_\pi\|_\xi}{1 - \sqrt{2\left(\alpha^2 + \vartheta^2\beta^2\right)}}. \qquad (19)$$

*In particular if $\beta_i = \beta = 0$ i.e., the proxy confidence region is actually the true confidence region, then the proxy projected Bellman equation has a solution satisfying $\|\widetilde{v}_\pi - v_\pi\|_\xi \leq \frac{\|\Pi v_\pi - v_\pi\|_\xi}{1-\alpha}$.*

Theorem 4.3 guarantees that the *robust projected Bellman iterations* of LSTD($\lambda$), LSPE($\lambda$) and TD($\lambda$)-methods converge, while Corollary 4.4 guarantees that the solution it coverges to is not too far away from the true value function $v_\pi$.

## 4.2 Robust approximations with nonlinear architectures

In this section we consider the situation where the function approximator $v_\theta$ is a smooth but not necessarily linear function of $\theta$. This section generalizes the results of [6] to the robust setting with confidence regions. We define robust analogues of the *nonlinear GTD2* and *nonlinear TDC* algorithms respectively.

Let $\mathcal{M} := \left\{ v_\theta \mid \theta \in \mathbb{R}^d \right\}$ be the manifold spanned by all possible value functions representable by our model and let $P\mathcal{M}_\theta$ be the *tangent plane* of $\mathcal{M}$ at $\theta$. Let $T\mathcal{M}_\theta$ be the *tangent space*, i.e., the translation of $P\mathcal{M}_\theta$ to the origin. In other words, $T\mathcal{M}_\theta := \left\{ \Phi_\theta u \mid u \in \mathbb{R}^d \right\}$, where $\Phi_\theta$ is an $n \times d$ matrix with entries $\Phi_\theta(i, j) := \frac{\partial}{\partial \theta_j} v_\theta(i)$. In the nonlinear case, we project on to the tangent space $T\mathcal{M}_\theta$, since projections on to $\mathcal{M}$ is computationally hard. We denote this projection by $\Pi_\theta$ and it is also with respect to a weighted Euclidean norm $\|\cdot\|_\xi$. The *mean squared projected Bellman equation* (MSPBE) loss function was proposed by [6] and is an extension of [22, 21], $\text{MSPBE}(\theta) = \|v_\theta - \Pi_\theta T_\pi v_\theta\|_\xi^2$, where we now project to the the tangent space $T\mathcal{M}_\theta$. Since the number $n$ of states is prohibitively large, we want stochastic gradient algorithms that run in time polynomial in $d$. Therefore, we assume that the confidence region of every state action pair is the same: $U_i^a = U$ and $\widehat{U}_i^a = U_i^a$. The robust version of the MSPBE loss function, the. *mean squared robust projected Bellman equation* (MSRPBE) loss can then be defined in terms of the *robust Bellman operator* with the proxy confidence region $\widehat{U}$ and proxy uncertainty set $\widehat{\mathcal{P}_i^{\pi(i)}}$ as

$$\text{MSRPBE}(\theta) = \left\| v_\theta - \Pi_\theta \widehat{T_\pi} v_\theta \right\|_\xi^2. \tag{20}$$

In order to derive stochastic gradient descent algorithms for minimizing the MSRPBE loss function, we need to take the gradient of $\sigma_P(v_\theta)$ for the a convex set $P$. The gradient $\mu$ of $\sigma$ is given by

$$\mu_P(\theta) := \nabla \max_{y \in P} y^\top v_\theta = \Phi_\theta^\top \arg\max_{y \in P} y^\top v_\theta, \tag{21}$$

where $\Phi_\theta(i) := \nabla v_\theta(i)$. Let us denote $\Phi_\theta(i)$ simply by $\phi$ and $\Phi_\theta(i')$ by $\phi'$, where $i'$ is the next sampled state. Let us denote by $\widehat{U}$ the proxy confidence region $\widehat{U_i^{\pi(i)}}$ of state $i$ and the policy $\pi$ under evaluation. Let

$$h(\theta, u) := -\mathbb{E}\left[ (\widetilde{d} - \phi^\top u)\nabla^2 v_\theta(i) u \right] \tag{22}$$

where $\widetilde{d}$ is the robust temporal difference error. As in [6], we may express $\nabla \text{MSRPBE}(\theta)$ in terms of $h(\theta, w)$ where $w = \mathbb{E}\left[\phi\phi^\top\right]^{-1} \mathbb{E}\left[\widetilde{d}\phi\right]$. We refer the reader to the supplementary material for the details. This leads us to the following robust analogues of *nonlinear GTD* and *nonlinear TDC*, where we update the estimators $w_k$ of $w$ as $w_{k+1} := w_k + \beta_k \left(\widetilde{d}_k - \phi_k^\top w_k\right)\phi_k$, with the parameters $\theta_k$ being updated on a slower timescale as

$$\theta_{k+1} := \Gamma\left(\theta_k + \alpha_k \left\{ \left(\phi_k - \vartheta\phi_k' - \vartheta\mu_{\widehat{U}}(\theta)\right)(\phi_k^\top w_k) - h_k\right\}\right) \qquad \text{robust-nonlinear-GTD2,} \tag{23}$$

$$\theta_{k+1} := \Gamma\left(\theta_k + \alpha_k \left\{ \widetilde{d}_k\phi_k - \vartheta\phi_k' - \vartheta\mu_{\widehat{U}}(\theta)(\phi_k^\top w_k) - h_k\right\}\right) \qquad \text{robust-nonlinear-TDC,} \tag{24}$$

where $h_k := \left(\widetilde{d}_k - \phi_k^\top w_k\right)\nabla^2 v_{\theta_k}(i_k) w_k$ and $\Gamma$ is a projection into an appropriately chosen compact set $C$ with a smooth boundary as in [6]. Under the assumption of Lipschitz continuous gradients

and suitable assumptions on the step lengths $\alpha_k$ and $\beta_k$ and the confidence region $\widehat{U}$, the updates of equations (23) converge with probability 1 to a local optima of MSRPBE$(\theta)$. See the supplementary material for the exact statement and proof of convergence. Note that in general computing $\mu_{\widehat{U}}(\theta)$ would take time polynomial in $n$, but it can be done in $O(d^2)$ time using a rank-$d$ approximation to $\widehat{U}$.

## 5 Experiments

We implemented robust versions of Q-learning and SARSA as in Section 3 and evaluated its performance against the nominal algorithms using the OpenAI gym framework [9]. To test the performance of the robust algorithms, we perturb the models slightly by choosing with a small probability $p$ a random state after every action. The size of the confidence region $U_i^a$ for the robust model is chosen by a 10-fold cross validation via line search. After the value functions are learned for the robust and the nominal algorithms, we evaluate its performance on the true environment. To compare the true algorithms we compare both the *cumulative reward* as well as the *tail distribution function* (complementary cumulative distribution function) as in [24] which for every $a$ plots the probability that the algorithm earned a reward of at least $a$.

Note that there is a tradeoff in the performance of the robust versus the nominal algorithms with the value of $p$ due to the presence of the $\beta$ term in the convergence results. See Figure 1 for a comparison. More figures and detailed results are included in the supplementary material.

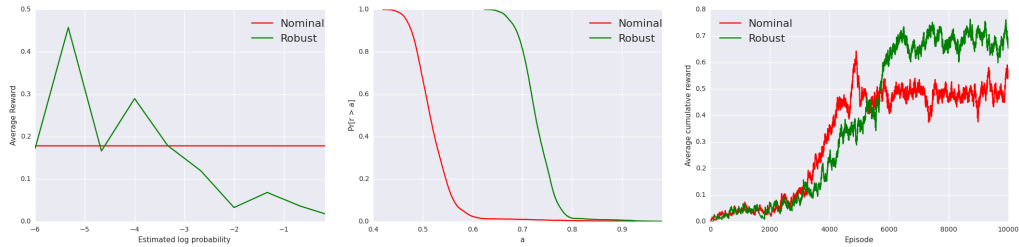

Figure 1: Line search, tail distribution, and cumulative rewards during transient phase of robust vs nominal Q-learning on **FrozenLake-v0** with $p = 0.01$. Note the instability of reward as a function of the size of the uncertainty set (left) is due to the small sample size used in line search.

### Acknowledgments

The authors would like to thank Guy Tennenholtz and anonymous reviewers for helping improve the presentation of the paper.

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
