[Supplementary Material · supplementary.pdf]

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

Figure 1: Example transition matrices shown within the probability simplex $\Delta_n$ with uncertainty sets being $\ell_2$ balls of fixed radius.

training. Note that while it may easy to come up with good descriptions of the confidence region $U_i^a$, the approach of [17, 13] breaks down since we have no knowledge of $p_i^a$ and merely observe the new state $j$ sampled from this distribution. See Figure 1 for an illustration with the confidence regions being an $\ell_2$ ball of fixed radius $r$.

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

*Proof.* Note that every point $p$ in $\mathcal{P}_i^a$ is of the form $p_i^a + x$ for some $x \in U_i^a$ and every point $q \in \widehat{\mathcal{P}_i^a}$ is of the form $p_i^a + y$ for some $y \in \widehat{U_i^a}$, and this correspondence is one to one by definition. For any

vector $v \in \mathbb{R}^n$ and pairs of points $p \in \mathcal{P}_i^a$ and $q \in \widehat{\mathcal{P}_i^a}$ we have

$$q^\top v = p^\top v + (q - p)^\top v \tag{11}$$

$$\leq \sup_{p' \in \mathcal{P}_i^a} (p')^\top v + (p_i^a + y - p_i^a - x)^\top v \tag{12}$$

$$= \sigma_{\mathcal{P}_i^a}(v) + (y - x)^\top v. \tag{13}$$

$$\leq \sigma_{\mathcal{P}_i^a}(v) + (y - x)^\top v \tag{14}$$

$$\leq \sigma_{\mathcal{P}_i^a}(v) + \left( y^\top v - \min_{x \in U_i^a} x^\top v \right) \tag{15}$$

$$\leq \sigma_{\mathcal{P}_i^a}(v) + \max_{y \in \widehat{U_i^a}} \min_{x \in U_i^a} (y - x)^\top v \tag{16}$$

$$\leq \sigma_{\mathcal{P}_i^a}(v) + \max_{y \in \widehat{U_i^a}} \min_{x \in U_i^a} \|y - x\|_1 \|v\|_\infty \tag{17}$$

$$\leq \sigma_{\mathcal{P}_i^a}(v) + \beta_i^a \|v\|_\infty . \tag{18}$$

Since equation (18) holds for every $q \in \widehat{\mathcal{P}_i^a}$, it follows that it also holds for $\arg\max \sigma_{\widehat{\mathcal{P}_i^a}}(v)$ so that

$$\sigma_{\widehat{\mathcal{P}_i^a}}(v) \leq \sigma_{\mathcal{P}_i^a}(v) + \beta_i^a \|v\|_\infty . \tag{19}$$

$\square$

The following theorem proves that under a suitable choice of step lengths $\gamma_t$ and discount factor $\vartheta$, the iteration of equation (9) converges to an $\varepsilon$-approximately optimal policy with respect to the confidence regions $U_i^a$.

**Theorem 3.3.** *Let the step lengths $\gamma_t$ of the Q-iteration algorithm be chosen such that $\sum_{t=0}^\infty \gamma_t = \infty$ and $\sum_{t=0}^\infty \gamma_t^2 < \infty$ and let the discount factor $\vartheta < 1$. Let $\beta_i^a$ be as in Lemma 3.2 and let $\beta :=$ $\max_{i \in \mathcal{X}, a \in \mathcal{A}} \beta_i^a$. If $\vartheta(1 + \beta) < 1$ then with probability $1$ the iteration of equation (9) converges to an $\varepsilon$-optimal policy where $\varepsilon := \frac{\vartheta\beta}{1 - \vartheta(1+\beta)}$.*

*Proof.* Let $\widehat{\mathcal{P}_i^a}$ be the proxy uncertainty set for state $i \in \mathcal{X}$ and $a \in \mathcal{A}$, i.e., $\widehat{\mathcal{P}_i^a} := \left\{ x + p_i^a \mid x \in \widehat{U_i^a} \right\}$. We denote the value function of Q by $v$. Let us define the following operator $H$ mapping Q-factors to Q-factors as follows:

$$(HQ)(i, a) := c(i, a) + \vartheta\sigma_{\widehat{\mathcal{P}_i^a}}(v). \tag{20}$$

We will first show that a solution $Q'$ to the equation $HQ = Q$ is an $\varepsilon$-optimal policy as in Definition 3.1, i.e., $\|Q' - Q^*\|_\infty \leq \varepsilon \|Q^*\|_\infty$.

$$\left| Q'(i,a) - Q^*(i,a) \right| = \left| (HQ')(i,a) - c(i,a) - \vartheta \sigma_{\mathcal{P}_i^a}(v^*) \right| \tag{21}$$

$$= \vartheta \left| \sigma_{\widehat{\mathcal{P}}_i^a}(v') - \sigma_{\mathcal{P}_i^a}(v^*) \right| \tag{22}$$

$$\leq \vartheta \left| \max_{y \in \widehat{U}_i^a, x \in U_i^a} \|y - x\|_1 \|Q'\|_\infty + \sigma_{\mathcal{P}_i^a}(v') - \sigma_{\mathcal{P}_i^a}(v^*) \right| \tag{23}$$

$$\leq \vartheta \beta_i^a \|Q'\|_\infty + \left| \sigma_{\mathcal{P}_i^a}(v') - \sigma_{\mathcal{P}_i^a}(v^*) \right| \tag{24}$$

$$\leq \vartheta \beta \|Q'\|_\infty + \vartheta \left| \max_{q' \in \mathcal{P}_i^a} \sum_{j \in \mathcal{X}} q'_j \min_{a'' \in \mathcal{A}} Q'(j,a'') - \max_{q \in \mathcal{P}_i^a} \sum_{j \in \mathcal{X}} q_j \min_{a' \in \mathcal{A}} Q^*(j,a') \right| \tag{25}$$

$$\leq \vartheta \beta \|Q'\|_\infty + \vartheta \left| \max_{q \in \mathcal{P}_i^a} \sum_{j \in \mathcal{X}} q_j \left( \min_{a'' \in \mathcal{A}} Q'(j,a'') - \min_{a' \in \mathcal{A}} Q^*(j,a') \right) \right| \tag{26}$$

$$\leq \vartheta \beta \|Q'\|_\infty + \vartheta \left| \max_{q \in \mathcal{P}_i^a} \sum_{j \in \mathcal{X}} q_j \left( \max_{a' \in \mathcal{A}} |Q'(j,a') - Q^*(j,a')| \right) \right| \tag{27}$$

$$\leq \vartheta \beta \|Q'\|_\infty + \vartheta \left| \max_{q \in \mathcal{P}_i^a} \sum_{j \in \mathcal{X}} q_j \|Q' - Q^*\|_\infty \right| \tag{28}$$

$$\leq \vartheta \beta \|Q'\|_\infty + \vartheta \|Q' - Q^*\|_\infty , \tag{29}$$

where we used Lemma 3.2 to derive equation (23). Equation (29) implies that $\|Q' - Q^*\|_\infty \leq \frac{\vartheta \beta}{1 - \vartheta} \|Q'\|_\infty$. If $\|Q'\|_\infty \leq \|Q^*\|_\infty$ then we are done since $\frac{\vartheta \beta}{1 - \vartheta} \leq \frac{\vartheta \beta}{1 - \vartheta(1 + \beta)}$. Otherwise assume that $\|Q'\|_\infty > \|Q^*\|_\infty$ and use the triangle inequality: $\|Q'\|_\infty - \|Q^*\|_\infty = \left| \|Q'\|_\infty - \|Q^*\|_\infty \right| \leq \|Q' - Q^*\|_\infty$. This implies that

$$\frac{1 - \vartheta}{\vartheta \beta} \|Q' - Q^*\|_\infty - \|Q^*\|_\infty \leq \|Q' - Q^*\|_\infty , \tag{30}$$

from which it follows that $\|Q' - Q^*\|_\infty \leq \varepsilon \|Q^*\|_\infty$ under the assumption that $\vartheta(1 + \beta) < 1$ as claimed. The Q-iteration of equation (9) can then be reformulated in terms of the operator $H$ as

$$Q_t(i,a) = (1 - \gamma_t) Q_{t-1}(i,a) + \gamma_t (H Q_t(i,a) + \eta_t(i,a)), \tag{31}$$

where $\eta_t(i,a) := \min_{a' \in \mathcal{A}} Q_t(j,a') - \mathbb{E}_{j \sim p_i^a}[\min_{a' \in \mathcal{A}} Q_t(j,a')]$ where the expectation is over the states $j \in \mathcal{X}$ with the transition probability from state $i$ to state $j$ given by $p_j^a$. Note that this is an example of a *stochastic approximation algorithm* as in [5] with noise parameter $\eta_t$. Let $\mathcal{F}_t$ denote the history of the algorithm until time $t$. Note that $\mathbb{E}_{j \sim p_i^a}[\eta_t(i,a)|\mathcal{F}_t] = 0$ by definition and the variance is bounded by

$$\mathbb{E}_{j \sim p_i^a} \left[ \eta_t(i,a)^2 \Big| \mathcal{F}_t \right] \leq K \left( 1 + \max_{\substack{j \in \mathcal{X} \\ a' \in \mathcal{A}}} Q_t^2(j,a') \right). \tag{32}$$

Thus the noise term $\eta_t$ satisfies the zero conditional mean and bounded variance assumption (Assumption 4.3 in [5]). Therefore it remains to show that the operator $H$ is a *contraction mapping* to argue that iterating equation (9) converges to the optimal Q-factor $Q^*$. We will show that the operator $H$ is a contraction mapping with respect to the infinity norm $\|.\|_\infty$. Let $Q$ and $Q'$ be two different Q-vectors with value functions $v$ and $v'$. If $U_i^a$ is not necessarily the same as the unconstrained proxy set $\widehat{U}_i^a$ for some $i \in \mathcal{X}, a \in \mathcal{A}$, then we need the discount factor to satisfy $\vartheta(1 + \beta)$ in order to ensure convergence. Intuitively, the discount factor should be small enough that the difference in the

estimation due to the difference of the sets $U_i^a$ and $\widehat{U}_i^a$ converges to 0 over time. In this case we show contraction for operator $H$ as follows

$$|(H\,Q)(i,a) - (H\,Q')(i,a)| \leq \vartheta \left| \max_{q \in \widehat{\mathcal{P}}_i^a} \sum_{j \in \mathcal{X}} q_j \left( \min_{a' \in \mathcal{A}} Q(j,a') - \min_{a'' \in \mathcal{A}} Q'(j,a'') \right) \right| \tag{33}$$

$$\leq \vartheta \max_{q \in \widehat{\mathcal{P}}_i^a} \sum_{j \in \mathcal{X}} q_j \max_{a' \in \mathcal{A}} |Q(j,a') - Q'(j,a')| \tag{34}$$

$$\leq \vartheta \max_{y \in \widehat{U}, x \in U} \|y - x\|_1 \|Q - Q'\|_\infty + \vartheta \max_{q \in \mathcal{P}_i^a} \sum_{j \in \mathcal{X}} q_j \|Q - Q'\|_\infty \tag{35}$$

$$\leq \vartheta \beta \|Q - Q'\|_\infty + \vartheta \|Q - Q'\|_\infty \max_{q \in \mathcal{P}_i^a} \sum_{j \in \mathcal{X}} q_j \tag{36}$$

$$\leq \vartheta(\beta + 1) \|Q - Q'\|_\infty \tag{37}$$

where we used Lemma 3.2 with vector $v(j) := \max_{a \in \mathcal{A}} |Q(j,a) - Q'(j,a)|$ to derive equation (35) and the fact that $\mathcal{P}_i^a \subseteq \Delta_n$ to conclude that $\max_{q \in \mathcal{P}_i^a} \sum_{j \in \mathcal{X}} q_j = 1$. Therefore if $\vartheta(1 + \beta) < 1$, then it follows that the operator $H$ is a norm contraction and thus the robust Q-iteration of equation (9) converges to a solution of $H\,Q = Q$ which is an $\varepsilon$-approximately optimal policy for $\varepsilon = \frac{\vartheta \beta}{1 - \vartheta(1+\beta)}$, as was proved before. $\square$

**Remark 3.4.** *If $\beta = 0$ then note that by Theorem 3.3, the robust Q-iterations converge to the exact optimal Q-factors since $\varepsilon = 0$. Since $\beta = \max_{i \in \mathcal{X}, a \in \mathcal{A}} \max_{y \in \widehat{U}_i^a} \min_{x \in U_i^a} \|y - x\|_1$, it follows that $\beta = 0$ iff $\widehat{U}_i^a = U_i^a$ for every $i \in \mathcal{X}, a \in \mathcal{A}$. This happens when the confidence region is small enough so that the simplex constraints $-p_{ij}^a \leq x_j \leq 1 - p_{ij}^a \forall j \in \mathcal{X}$ in the description of $\mathcal{P}_i^a$ become redundant for every $i \in \mathcal{X}, a \in \mathcal{A}$. Equivalently every $p_i^a$ is "far" from the boundary of the simplex $\Delta_n$ compared to the size of the confidence region $U_i^a$, see e.g., Figure 1.*

**Remark 3.5.** *Note that simply using the nominal Q-iteration without the $\sigma_{\widehat{U}_i^a}(v)$ term does not guarantee convergence to $Q^*$. Indeed, the nominal Q-iterations converge to Q-factors $Q'$ where $\|Q' - Q^*\|_\infty$ may be arbitrary large. This follows easily from observing that $|Q'(i,a) - Q^*(i,a)| = \left| \sigma_{\widehat{U}_i^a}(v^*) \right|$, where $v^*$ is the value function of $Q^*$ and so*

$$\|Q' - Q^*\|_\infty = \max_{i \in \mathcal{X}, a \in \mathcal{A}} \left| \sigma_{\widehat{U}_i^a}(v^*) \right|, \tag{38}$$

*which can be as high as $\|v^*\|_\infty = \|Q^*\|_\infty$. See Section 5 for an experimental demonstration of the difference in the policies learned by the robust and nominal algorithms.*

### 3.2 Robust SARSA

Recall that the update rule of SARSA is similar to the update rule for Q-learning except that instead of choosing the action $a' = \arg\min_{a' \in \mathcal{A}} Q_{t-1}(j,a')$, we choose the action $a''$ where with probability $\delta$, the action $a''$ is chosen uniformly at random from $\mathcal{A}$ and with probability $1 - \delta$, we have $a'' = \arg\min_{a' \in \mathcal{A}} Q_{t-1}(j,a')$. Therefore, it is easy to modify the robust Q-iteration of equation (9) to give us the *robust* SARSA updates:

$$Q_t(i,a) := (1 - \gamma_t) Q_{t-1}(i,a) + \gamma_t \left( c(i,a) + \vartheta \sigma_{\widehat{U}_i^a}(v_{t-1}) + \vartheta Q_{t-1}(j,a'') \right). \tag{39}$$

In the exact dynamic programming setting, it has the same convergence guarantees as robust Q-learning and can be seen as a corollary of Theorem 3.3.

**Corollary 3.6.** *Let the step lengths $\gamma_t$ be chosen such that $\sum_{t=0}^\infty \gamma_t = \infty$ and $\sum_{t=0}^\infty \gamma_t^2 < \infty$ and let the discount factor $\vartheta < 1$. Let $\beta_i^a$ be as in Lemma 3.2 and let $\beta := \max_{i \in \mathcal{X}, a \in \mathcal{A}} \beta_i^a$. If $\vartheta(1 + \beta) < 1$ then with probability 1 the iteration of equation (39) converges to an $\varepsilon$-optimal policy where $\varepsilon := \frac{\vartheta \beta}{1 - \vartheta(1+\beta)}$. In particular if $\beta = \beta_i^a = 0$ so that the proxy confidence regions $\widehat{U}_i^a$ are the same as the true confidence regions $U_i^a$, then the iteration (39) converges to the true optimum $Q^*$.*

### 3.3 Robust TD-learning

Recall that TD-learning allows us to estimate the value function $v_\pi$ for a given policy $\pi$. In this section we will generalize the TD-learning algorithm to the robust case. The main idea behind TD-learning in the non-robust setting is the following Bellman equation

$$v_\pi(i) := \mathbb{E}_{j \sim p_i^{\pi(i)}} \left[ c(i, \pi(i)) + v_\pi(j) \right]. \tag{40}$$

Consider a trajectory of the agent $(i_0, i_1, \dots)$, where $i_m$ denotes the state of the agent at time step $m$. For a time step $m$, define the *temporal difference* $d_m$ as

$$d_m := c(i_m, \pi(i_m)) + \vartheta v_\pi(i_{m+1}) - v_\pi(i_m). \tag{41}$$

Let $\lambda \in (0, 1)$. The recurrence relation for $TD(\lambda)$ may be written in terms of the temporal difference $d_m$ as

$$v_\pi(i_k) = \mathbb{E} \left[ \sum_{m=0}^{\infty} (\vartheta\lambda)^{m-k} d_m \right] + v_\pi(i_k). \tag{42}$$

The corresponding Robbins-Monro stochastic approximation algorithm with step size $\gamma_t$ for equation (42) is

$$v_{t+1}(i_k) := v_t(i_k) + \gamma_t \left( \sum_{m=k}^{\infty} (\vartheta\lambda)^{m-k} d_m \right). \tag{43}$$

A more general variant of the $TD(\lambda)$ iterations uses *eligibility coefficients* $z_m(i)$ for every state $i \in \mathcal{X}$ and temporal difference vector $d_m$ in the update for equation (43)

$$v_{t+1}(i) := v_t(i) + \gamma_t \left( \sum_{m=k}^{\infty} z_m(i) d_m \right). \tag{44}$$

Let $i_m$ denote the state of the simulator at time step $m$. For the discounted case, there are two possibilities for the eligibility vectors $z_m(i)$ leading to two different $TD(\lambda)$ iterations:

1. The *every-visit* $TD(\lambda)$ method, where the eligibility coefficients are

$$z_m(i) := \begin{cases} \vartheta\lambda z_{m-1}(i) & \text{if } i_m \neq i \\ \vartheta\lambda z_{m-1}(i) + 1 & \text{if } i_m = i. \end{cases}$$

2. The *restart* $TD(\lambda)$ method, where the eligibility coefficients are

$$z_m(i) := \begin{cases} \vartheta\lambda z_{m-1}(i) & \text{if } i_m \neq i \\ 1 & \text{if } i_m = i. \end{cases}$$

We make the following assumptions about the eligibility coefficients that are sufficient for proof of convergence.

**Assumption 3.7.** *The eligibility coefficients $z_m$ satisfy the following conditions*

1. *$z_m(i) \geq 0$*

2. *$z_{-1}(i) = 0$*

3. *$z_m(i) \leq \vartheta z_{m-1}(i)$ if $i \notin \{i_0, i_1, \dots\}$*

4. *The weight $z_m(i)$ given to the temporal difference $d_m$ should be chosen before this temporal difference is generated.*

Note that the eligibility coefficients of both the every-visit and restart $TD(\lambda)$ iterations satisfy Assumption 3.7. In the robust setting, we are interested in estimating the *robust value* of a policy $\pi$, which from Theorem 2.2 we may express as

$$v_\pi(i) := c(i, \pi(i)) + \vartheta \max_{p \in \mathcal{P}_i^{\pi(i)}} \mathbb{E}_{j \sim p} \left[ v_\pi(j) \right], \tag{45}$$

where the expectation is now computed over the probability vector $p$ chosen adversarially from the uncertainty region $\mathcal{P}_i^a$. As in Section 3.1, we may decompose $\max_{p \in \mathcal{P}_i^a} \mathbb{E}_{j \sim p}\left[v(j)\right] = \sigma_{\mathcal{P}_i^a}(v)$ as

$$\max_{p \in \mathcal{P}_i^{\pi(i)}} \mathbb{E}_{j \sim p}\left[v(j)\right] = \sigma_{U_i^{\pi(i)}}(v) + \mathbb{E}_{j \sim p_i^{\pi(i)}}\left[v(j)\right], \tag{46}$$

where $p_i^{\pi(i)}$ is the transition probability of the agent during a simulation. For the remainder of this section, we will drop the subscript and just use $\mathbb{E}$ to denote expectation with respect to this transition probability $p_i^{\pi(i)}$.

Define a *simulation* to be a trajectory $\{i_0, i_1, \ldots, i_{N_t}\}$ of the agent, which is stopped according to a random *stopping time* $N_t$. Note that $N_t$ is a random variable for making stopping decisions that is not allowed to foresee the future. Let $\mathcal{F}_t$ denote the history of the algorithm up to the point where the $t^{th}$ simulation is about to commence. Let $v_t$ be the estimate of the value function at the start of the $t^{th}$ simulation. Let $\{i_0, i_1, \ldots, i_{N_t}\}$ be the trajectory of the agent during the $t^{th}$ simulation with $i_0 = i$. During training, we generate several simulations of the agent and update the estimate of the *robust* value function using the the *robust temporal difference* $\widetilde{d}_m$ which is defined as

$$\widetilde{d}_m := d_m + \vartheta \sigma_{\widehat{U_{i_m}^{\pi(i_m)}}}(v_t), \tag{47}$$

$$= c(i_m, \pi(i_m)) + \vartheta v_t(i_{m+1}) - v_t(i_m) + \vartheta \sigma_{\widehat{U_{i_m}^{\pi(i_m)}}}(v_t), \tag{48}$$

where $d_m$ is the usual temporal difference defined as before

$$d_m := c(i_m, \pi(i_m)) + \vartheta v_t(i_{m+1}) - v_t(i_m). \tag{49}$$

The *robust* TD-update is now the usual TD-update, except that we use the *robust temporal difference* computed over the proxy confidence region:

$$v_{t+1}(i) := v_t(i) + \gamma_t \sum_{m=0}^{N_t-1} z_m(i)\left(\widetilde{d}_m\right), \tag{50}$$

$$= v_t(i) + \gamma_t \sum_{m=0}^{N_t-1} z_m(i)\left(\vartheta \sigma_{\widehat{U_{i_m}^{\pi(i_m)}}}(v_t) + d_m\right). \tag{51}$$

We define an $\varepsilon$-approximate value function for a fixed policy $\pi$ in a way similar to the $\varepsilon$-optimal Q-factors as in Definition 3.1:

**Definition 3.8** ($\varepsilon$-approximate value function). *Given a policy $\pi$, we say that a vector $v' \in \mathbb{R}^n$ is an $\varepsilon$-approximation of $v_\pi$ if the following holds*

$$\left\|v' - v_\pi\right\|_\infty \leq \varepsilon \left\|v_\pi\right\|_\infty.$$

The following theorem guarantees convergence of the robust TD iteration of equation (50) to an approximate value function for $\pi$ under Assumption 3.7.

**Theorem 3.9.** *Let $\beta_i^a$ be as in Lemma 3.2 and let $\beta := \max_{i \in \mathcal{X}, a \in \mathcal{A}} \beta_i^a$. Let $\rho := \max_{i \in \mathcal{X}} \sum_{m=0}^{\infty} z_m(i)$. If $\vartheta(1 + \rho\beta) < 1$ then the robust TD-iterations of equation (50) converges to an $\varepsilon$-approximate value function, where $\varepsilon := \frac{\vartheta\beta}{1 - \vartheta(1+\rho\beta)}$. In particular if $\beta_i^a = \beta = 0$, i.e., the proxy confidence region $\widehat{U_i^a}$ is the same as the true confidence region $U_i^a$, then the convergence is exact, i.e., $\varepsilon = 0$. Note that in the special case of regular $\mathrm{TD}(\lambda)$ iterations, $\rho = \frac{\vartheta\lambda}{1 - \vartheta\lambda}$.*

*Proof.* Let $\widehat{\mathcal{P}_i^a}$ be the proxy uncertainty set for state $i \in \mathcal{X}$ and action $a \in \mathcal{A}$ as in the proof of Theorem 3.3, i.e., $\widehat{\mathcal{P}_i^a} := \left\{x + p_i^a \mid x \in \widehat{U_i^a}\right\}$. Let $I_t(i) := \{m \mid i_m = i\}$ be the set of time indices the $t^{th}$ simulation visits state $i$. We define $\delta_t(i) := \max_{q_m \in \mathcal{P}_{i_m}^{\pi(i_m)}} \mathbb{E}_{i_m \sim q_m}\left[\sum_{m \in I_t(i)} z_m(i) \Big| \mathcal{F}_t\right]$, so

that we may write the update of equation (50) as

$$v_{t+1}(i) = v_t(i)(1 - \gamma_t\delta_t(i)) + \gamma_t\delta_t(i)\left(\frac{\mathbb{E}\left[\sum_{m=0}^{N_t-1} z_m(i)\widetilde{d}_m\Big|\mathcal{F}_t\right]}{\delta_t(i)} + v_t(i)\right) \tag{52}$$

$$+\gamma_t\delta_t(i)\frac{\vartheta\sum_{m=0}^{N_t-1} z_m(i)\widetilde{d}_m - \mathbb{E}\left[\sum_{m=0}^{N_t-1} z_m(i)\widetilde{d}_m\Big|\mathcal{F}_t\right]}{\delta_t(i)}. \tag{53}$$

Let us define the operator $H_t : \mathbb{R}^n \to \mathbb{R}^n$ corresponding to the $t^{th}$ simulation as

$$(H_t v)(i) := \frac{\mathbb{E}\left[\sum_{m=0}^{N_t-1} z_m(i)\left(c(i_m, \pi(i_m)) + \vartheta\sigma_{\widehat{U_{i_m}^{\pi(i_m)}}}(v) + \vartheta v(i_{m+1}) - v(i_m)\right)\Big|\mathcal{F}_t\right]}{\delta_t(i)} + v(i). \tag{54}$$

We claim as in the proof of Theorem 3.3 that a solution $v$ to $H_t v = v$ must be an $\varepsilon$-approximation to $v_\pi$. Define the operator $H_t'$ with the proxy confidence regions replaced by the true ones, i.e.,

$$(H_t' v)(i) := \frac{\mathbb{E}\left[\sum_{m=0}^{N_t-1} z_m(i)\left(c(i_m, \pi(i_m)) + \vartheta\sigma_{U_{i_m}^{\pi(i_m)}}(v) + \vartheta v(i_{m+1}) - v(i_m)\right)\Big|\mathcal{F}_t\right]}{\delta_t(i)} + v(i). \tag{55}$$

Note that $H_t' v_\pi = v_\pi$ for the *robust* value function $v_\pi$ since $c(i_m, \pi(i_m)) + \vartheta\sigma_{U_{i_m}^{\pi(i_m)}}(v_\pi) + \vartheta v_\pi(i_{m+1}) - v_\pi(i_m) = 0$ for every $i_m \in \mathcal{X}$ by Theorem 2.2. Finally by Lemma 3.2 we have

$$\sigma_{\widehat{U_{i_m}^{\pi(i_m)}}}(v) + \mathbb{E}\left[v(i_m)\right] \le \sigma_{U_{i_m}^{\pi(i_m)}} + \mathbb{E}\left[v(i_m)\right] + \beta\|v\|_\infty, \tag{56}$$

for any vector $v$, where the expectation is over the state $i_m \sim p_{i_{m-1}}^{\pi(i_{m-1})}$. Thus for any solution $v$ to the equation $H_t v = v$, we have

$$|v(i) - v_\pi(i)| = |(H_t v)(i) - v_\pi(i)| \tag{57}$$

$$\le |(H_t' v)(i) - v_\pi(i)| + \vartheta\beta\|v\|_\infty \mathbb{E}\left[\sum_{m=0}^{N_t-1} z_m(i)\right] \tag{58}$$

$$= |(H_t' v)(i) - (H_t' v_\pi)(i)| + \vartheta\beta\|v\|_\infty \mathbb{E}\left[\sum_{m=0}^{N_t-1} z_m(i)\right] \tag{59}$$

$$\le \vartheta\|v - v_\pi\|_\infty + \vartheta\rho\beta\|v\|_\infty, \tag{60}$$

where equation (60) follows from equation (55). Therefore the solution to $H_t v = v$ is an $\varepsilon$-approximation to $v_\pi$ for $\varepsilon = \frac{\vartheta\beta}{1-\vartheta(1+\rho\beta)}$ if $\vartheta(1 + \rho\beta) < 1$ as in the proof of Theorem 3.3. Note that the operator $H_t$ applied to the iterates $v_t$ is $(H_t v_t)(i) = \frac{\mathbb{E}\left[\sum_{m=0}^{N_t-1} z_m^t(i)\widetilde{d}_{m,t}|\mathcal{F}_t\right]}{\delta_t(i)} + v_t(i)$ so that the update of equation (50) is a *stochastic approximation algorithm* of the form

$$v_{t+1}(i) = (1 - \widehat{\gamma}_t)v_t(i) + \widehat{\gamma}_t\left((H_t v_t)(i) + \eta_t(i)\right),$$

where $\widehat{\gamma}_t = \gamma_t\delta_t(i)$ and $\eta_t$ is a noise term with zero mean and is defined as

$$\eta_t(i) := \frac{\sum_{m=0}^{N_t-1} z_m^t(i)\widetilde{d}_m - \mathbb{E}\left[\sum_{m=0}^{N_t-1} z_m^t(i)\widetilde{d}_m\Big|\mathcal{F}_t\right]}{\delta_t(i)}. \tag{61}$$

Note that by Lemma 5.1 of [5], the new step sizes satisfy $\sum_{t=0}^\infty \widehat{\gamma}_t = \infty$ and $\sum_{t=0}^\infty \widehat{\gamma}_t^2 < \infty$ if the original step size $\gamma_t$ satisfies the conditions $\sum_{t=0}^\infty \gamma_t = \infty$ and $\sum_{t=0}^\infty \gamma_t^2 < \infty$, since the conditions on the eligibility coefficients are unchanged. Note that the noise term also satisfies the bounded variance of Lemma 5.2 of [5] since any $q \in \mathcal{P}_i^{\pi(i)}$ still specifies a distribution as $\mathcal{P}_i^{\pi(i)} \subseteq \Delta_n$.

Therefore, it remains to show that $H_t$ is a norm contraction with respect to the $\ell_\infty$ norm on $v$. Let us define the operator $A_t$ as

$$(A_t v)(i) := \frac{\mathbb{E}\left[\sum_{m=0}^{N_t-1} z_m(i)\left(\vartheta\sigma_{\widehat{U_{i_m}^{\pi(i_m)}}}(v) + \vartheta v(i_{m+1}) - v(i_m)\,\middle|\,\mathcal{F}_t\right)\right]}{\delta_t(i)} + v(i) \qquad (62)$$

and the expression $b_t(i) := \frac{\mathbb{E}\left[\sum_{m=0}^{N_t-1} c(i_m, \pi(i_m))\,\middle|\,\mathcal{F}_t\right]}{\delta_t(i)}$ so that $(H_t v)(i) = (A_t v)(i) + b_t(i)$. We will show that $\|A_t v\|_\infty \le \alpha \|v\|_\infty$ for some $\alpha < 1$ from which the contraction on $H_t$ follows because for any vector $v'' \in \mathbb{R}^n$ and the $\varepsilon$-optimal value function $v' = H_t v'$ we have

$$\left\|H_t v'' - v'\right\|_\infty = \left\|H_t v'' - H_t v'\right\|_\infty = \left\|A_t(v'' - v')\right\|_\infty \le \alpha \left\|v'' - v'\right\|_\infty. \qquad (63)$$

Let us now analyze the expression for $A_t$. We will show that

$$\mathbb{E}\left[\sum_{m=0}^{N_t-1} z_m(i)\left(\vartheta v(i_{m+1}) - v(i_m) + \vartheta\sigma_{\widehat{U_i^{\pi(i)}}}(v)\right) + \sum_{m \in I_t(i)} z_m(i) v(i)\,\middle|\,\mathcal{F}_t\right] \le \qquad (64)$$

$$\alpha \|v\|_\infty \mathbb{E}\left[\sum_{m \in I_t(i)} z_m(i)\,\middle|\,\mathcal{F}_t\right]. \qquad (65)$$

We first replace the $\sigma_{\widehat{U_{i_m}^{\pi(i_m)}}}$ term with $\sigma_{U_{i_m}^{\pi(i_m)}}$ using Lemma 3.2 while incurring a $\rho\beta \|v\|_\infty$ penalty. Let us collect together the coefficients corresponding to $v(i_m)$ in the expression for the expectation:

$$\mathbb{E}\left[\sum_{m=0}^{N_t-1} z_m(i)\left(\vartheta v(i_{m+1}) - v(i_m) + \vartheta\sigma_{U_{i_m}^{\pi(i_m)}}(v)\right) + \sum_{m \in I_t(i)} z_m(i) v(i)\,\middle|\,\mathcal{F}_t\right] + \vartheta\rho\beta \|v\|_\infty \qquad (66)$$

$$\le \max_{q_m \in \mathcal{P}_{i_m}^{\pi(i_m)}} \mathbb{E}_{i_m \sim q_m}\left[\sum_{m=0}^{N_t-1} z_m(i)\left(\vartheta v(i_{m+1}) - v(i_m)\right) + \sum_{m \in I_t(i)} z_m(i) v(i)\,\middle|\,\mathcal{F}_t\right] + \vartheta\rho\beta \|v\|_\infty \qquad (67)$$

$$= \max_{q_m \in \mathcal{P}_{i_m}^{\pi(i_m)}} \mathbb{E}_{i_m \sim q_m}\left[\sum_{m=0}^{N_t}\left(\vartheta z_{m-1}(i) - z_m(i)\right) v(i_m) + \sum_{m \in I_t(i)} z_m(i) v(i)\,\middle|\,\mathcal{F}_t\right] + \vartheta\rho\beta \|v\|_\infty, \qquad (68)$$

where we obtain inequality (67) by subsuming the $\sigma_{U_{i_m}^{\pi(i_m)}}$ term within the expectation since $\mathcal{P}_{i_m}^{\pi(i_m)}$ is now part of the simplex $\Delta_n$ and taking the worst possible distribution $q_m$. We also used the fact that $z_{-1}(i) = 0$ and $z_{N_t}(i) = 0$. Note that whenever $i_m \ne i$, the coefficient $\vartheta z_{m-1}(i) - z_m(i)$ of $v(i_m)$ is nonnegative while whenever $i_m = i$, then the coefficient $\vartheta z_{m-1}(i) - z_m(i) + z_m(i)$ is also nonnegative. Therefore, we may bound the right hand side of equation (66) as

$$\max_{q_m \in \mathcal{P}_{i_m}^{\pi(i_m)}} \mathbb{E}_{i_m \sim q_m}\left[\sum_{m=0}^{N_t}\left(\vartheta z_{m-1}(i) - z_m(i)\right) v(i_m) + \sum_{m \in I_t(i)} z_m(i) v(i)\,\middle|\,\mathcal{F}_t\right] + \vartheta\rho\beta \|v\|_\infty \qquad (69)$$

$$\le \max_{q_m \in \mathcal{P}_{i_m}^{\pi(i_m)}} \mathbb{E}_{i_m \sim q_m}\left[\sum_{m=0}^{N_t}\left(\vartheta z_{m-1}(i) - z_m(i)\right) \|v\|_\infty + \sum_{m \in I_t(i)} z_m(i) \|v\|_\infty\,\middle|\,\mathcal{F}_t\right] + \vartheta\rho\beta \|v\|_\infty. \qquad (70)$$

Let us now collect the terms corresponding to a fixed $z_m(i)$:

$$\max_{q_m \in \mathcal{P}_{i_m}^{\pi(i_m)}} \mathbb{E}_{i_m \sim q_m} \left[ \sum_{m=0}^{N_t} (\vartheta z_{m-1}(i) - z_m(i)) \|v\|_\infty + \sum_{m \in I_t(i)} z_m(i) \|v\|_\infty \Bigg| \mathcal{F}_t \right] + \vartheta \rho \beta \|v\|_\infty \quad (71)$$

$$= \|v\|_\infty \max_{q_m \in \mathcal{P}_{i_m}^{\pi(i_m)}} \mathbb{E}_{i_m \sim q_m} \left[ \sum_{m=0}^{N_t-1} z_m(i) (\vartheta - 1) + \sum_{m \in I_t(i)} z_m(i) \Bigg| \mathcal{F}_t \right] + \vartheta \rho \beta \|v\|_\infty \quad (72)$$

$$\leq \|v\|_\infty \max_{q_m \in \mathcal{P}_{i_m}^{\pi(i_m)}} \mathbb{E}_{i_m \sim q_m} \left[ \sum_{m \in I_t(i)} z_m(i) (\vartheta - 1) + \sum_{m \in I_t(i)} z_m(i) \Bigg| \mathcal{F}_t \right] + \vartheta \rho \beta \|v\|_\infty \quad (73)$$

$$\leq \|v\|_\infty \vartheta (1 + \rho \beta) \mathbb{E} \left[ \sum_{m \in I_t(i)} z_m(i) \Bigg| \mathcal{F}_t \right] \quad (74)$$

where equation (73) follows since $\vartheta < 1$. Therefore setting $\alpha = \vartheta (1 + \rho \beta)$, our claim follows under the assumption that $\vartheta(1 + \rho \beta) < 1$. $\qquad\square$

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

*Proof.* Same as Lemma 3.2 except now we take Cauchy-Schwarz with respect to weighted Euclidean norm $\|\cdot\|_\xi$ in the following manner

$$a^\top b \leq \frac{a^\top \Xi b}{\xi_{\min}} \leq \frac{\|a\|_\xi \|b\|_\xi}{\xi_{\min}}. \tag{78}$$

$\square$

The following theorem shows that the robust projected Bellman equation is a contraction under reasonable assumptions on the discount factor $\vartheta$.

**Theorem 4.3.** *Let $\beta_i^a$ be as in Lemma 4.2 and let $\beta := \max_{i \in \mathcal{X}} \beta_i^{\pi(i)}$. If the discount factor $\vartheta$ satisfies Assumption 4.1 for some $\alpha$ and $\alpha^2 + \vartheta^2\beta^2 < \frac{1}{2}$, then the operator $\widehat{T_\pi}$ is a contraction with respect to $\|\cdot\|_\xi$. In other words, for any two $\theta, \theta' \in \mathbb{R}^d$, we have*

$$\left\|\widehat{T_\pi}(\Phi\theta) - \widehat{T_\pi}(\Phi\theta')\right\|_\xi^2 \leq 2\left(\alpha^2 + \vartheta^2\beta^2\right) \|\Phi\theta - \Phi\theta'\|_\xi^2 < \|\Phi\theta - \Phi\theta'\|_\xi^2. \tag{79}$$

*If $\beta_i = \beta = 0$ so that $\widehat{U_i^{\pi(i)}} = U_i^{\pi(i)}$, then we have a simpler contraction under the assumption that $\alpha < 1$, i.e.,*

$$\left\|\widehat{T_\pi}(\Phi\theta) - \widehat{T_\pi}(\Phi\theta')\right\|_\xi \leq \alpha \|\Phi\theta - \Phi\theta'\|_\xi < \|\Phi\theta - \Phi\theta'\|_\xi. \tag{80}$$

*Proof.* Consider two parameters $\theta$ and $\theta'$ in $\mathbb{R}^d$. Then we have

$$\left\|\widehat{T}_\pi(\Phi^\top \theta) - \widehat{T}_\pi(\Phi^\top \theta')\right\|_\xi^2 = \sum_{i \in \mathcal{X}} \xi_i \left(\widehat{T}_\pi(\Phi^\top \theta)(i) - \widehat{T}_\pi(\Phi^\top \theta')(i)\right)^2 \tag{81}$$

$$= \vartheta^2 \sum_{i \in \mathcal{X}} \xi_i \left(\sigma_{\Phi^\top\left(\widehat{\mathcal{P}_i^{\pi(i)}}\right)}(\theta) - \sigma_{\Phi^\top\left(\widehat{\mathcal{P}_i^{\pi(i)}}\right)}(\theta')\right)^2 \tag{82}$$

$$= \vartheta^2 \sum_{i \in \mathcal{X}} \xi_i \left(\sup_{q \in \widehat{\mathcal{P}_i^{\pi(i)}}} q^\top \Phi\theta - \sup_{q' \in \widehat{\mathcal{P}_i^{\pi(i)}}} (q')^\top \Phi\theta'\right)^2 \tag{83}$$

$$\leq \vartheta^2 \sum_{i \in \mathcal{X}} \xi_i \left(\sup_{q \in \widehat{\mathcal{P}_i^{\pi(i)}}} q^\top \left(\Phi\theta - \Phi\theta'\right)\right)^2 \tag{84}$$

$$\leq \vartheta^2 \sum_{i \in \mathcal{X}} \xi_i \left(\sup_{q \in \mathcal{P}_i^{\pi(i)}} \left(q^\top \left(\Phi\theta - \Phi\theta'\right)\right) + \beta \left\|\Phi\theta - \Phi\theta'\right\|_\xi\right)^2 \tag{85}$$

$$\leq \sum_{i \in \mathcal{X}} \xi_i \left(\alpha \sum_{j \in \mathcal{X}} P_{ij}^{\widehat{\pi}} \left(\phi(j)^\top \theta - \phi(j)^\top \theta'\right) + \vartheta\beta \left\|\Phi\theta - \Phi\theta'\right\|_\xi\right)^2 \tag{86}$$

$$\leq 2 \sum_{i \in \mathcal{X}} \xi_i \left(\alpha^2 \sum_{j \in \mathcal{X}} P_{ij}^{\widehat{\pi}} \left(\phi(j)^\top \theta - \phi(j)^\top \theta'\right)^2 + \vartheta^2\beta^2 \left\|\Phi\theta - \Phi\theta'\right\|_\xi^2\right) \tag{87}$$

$$\leq 2(\alpha^2 + \vartheta^2\beta^2) \left\|\Phi\theta - \Phi\theta'\right\|_\xi^2 \tag{88}$$

where we used Lemma 4.2 and the definition of $\beta$ in line (85), the inequality $(a+b)^2 \leq 2(a^2+b^2)$, and the fact that $\left(P_{ij}^{\widehat{\pi}}\right)^2 \leq P_{ij}^{\widehat{\pi}}$. Note that if $\beta_i^{\pi(i)} = \beta = 0$ so that the proxy confidence region is the same as the true confidence region, then we have the simple upper bound of $\left\|\widehat{T}_\pi(\Phi^\top \theta) - \widehat{T}_\pi(\Phi^\top \theta')\right\|_\xi^2 \leq \alpha^2 \left\|\Phi\theta - \Phi\theta'\right\|_\xi^2$ instead of $\left\|\widehat{T}_\pi(\Phi^\top \theta) - \widehat{T}_\pi(\Phi^\top \theta')\right\|_\xi^2 \leq 2\alpha^2 \left\|\Phi\theta - \Phi\theta'\right\|_\xi^2$ since we do not have the cross term in equation (86) in this case. □

The following corollary shows that the solution to the proxy projected Bellman equation converges to a solution that is not too far away from the true value function $v_\pi$.

**Corollary 4.4.** *Let Assumption 4.1 hold and let $\beta$ be as in Theorem 4.3. Let $\widetilde{v}_\pi$ be the fixed point of the projected Bellman equation for the proxy operator $\widehat{T}_\pi$, i.e., $\Pi\widehat{T}_\pi\widetilde{v}_\pi = \widetilde{v}_\pi$. Let $\widehat{v}_\pi$ be the fixed point of the proxy operator $\widehat{T}_\pi$, i.e., $\widehat{T}_\pi\widehat{v}_\pi = \widehat{v}_\pi$. Let $v_\pi$ be the true value function of the policy $\pi$, i.e., $T_\pi v_\pi = v_\pi$. Then the following holds*

$$\|\widetilde{v}_\pi - v_\pi\|_\xi \leq \frac{\vartheta\beta \|v_\pi\|_\xi + \|\Pi v_\pi - v_\pi\|_\xi}{1 - \sqrt{2\left(\alpha^2 + \vartheta^2\beta^2\right)}}. \tag{89}$$

*In particular if $\beta_i = \beta = 0$ i.e., the proxy confidence region is actually the true confidence region, then the proxy projected Bellman equation has a solution satisfying $\|\widetilde{v}_\pi - v_\pi\|_\xi \leq \frac{\|\Pi v_\pi - v_\pi\|_\xi}{1-\alpha}$.*

*Proof.* We have the following expression

$$\|\widetilde{v}_\pi - v_\pi\|_\xi \leq \|\widetilde{v}_\pi - \Pi v_\pi\|_\xi + \|\Pi v_\pi - v_\pi\|_\xi \tag{90}$$

$$\leq \left\|\Pi\widehat{T}_\pi\widetilde{v}_\pi - \Pi T_\pi v_\pi\right\|_\xi + \|\Pi v_\pi - v_\pi\|_\xi \tag{91}$$

$$\leq \left\|\Pi\widehat{T}_\pi\widetilde{v}_\pi - \Pi\widehat{T}_\pi v_\pi + \vartheta\beta\|v_\pi\|_\xi\right\| + \|\Pi v_\pi - v_\pi\|_\xi \tag{92}$$

$$\leq \left\|\Pi\widehat{T}_\pi\widetilde{v}_\pi - \Pi\widehat{T}_\pi v_\pi\right\|_\xi + \vartheta\beta\|v_\pi\|_\xi + \|\Pi v_\pi - v_\pi\|_\xi \tag{93}$$

$$\leq \sqrt{2(\alpha^2 + \vartheta^2\beta^2)}\|\widetilde{v}_\pi - v_\pi\|_\xi + \vartheta\beta\|v_\pi\|_\xi + \|\Pi v_\pi - v_\pi\|_\xi, \tag{94}$$

where we used Lemma 4.2 to derive inequality (92) and Theorem 4.3 to conclude that $\left\|\Pi\widehat{T}_\pi\widetilde{v}_\pi - \Pi\widehat{T}_\pi v_\pi\right\|_\xi \leq \sqrt{2(\alpha^2 + \vartheta^2\beta^2)}\|\widetilde{v}_\pi - v_\pi\|_\xi$. If $\beta_i^{\pi(i)} = \beta = 0$ so that the proxy confidence regions are the same as the true confidence regions, then we have $\alpha$ instead of $\sqrt{2(\alpha^2 + \vartheta^2\beta^2)}$ in the last equation due to Theorem 4.3. $\square$

Theorem 4.3 guarantees that the *robust projected Bellman iterations* of LSTD($\lambda$), LSPE($\lambda$) and TD($\lambda$)-methods converge, while Corollary 4.4 guarantees that the solution it converges to is not too far away from the true value function $v_\pi$. We refer the reader to [3] for more details on LSTD($\lambda$), LSPE($\lambda$) since their proof of convergence is analogous to that of TD($\lambda$).

## 4.2 Robust stochastic gradient descent algorithms

While the TD($\lambda$)-learning algorithms with function approximation with linear architectures converges to $v_\pi$ if the states are sampled according to the policy $\pi$, it is known to be unstable if the states are sampled in an *off-policy* manner, i.e., in the terminology of the previous section $\widehat{\pi} \neq \pi$. This issue was addressed by [24, 23] who proposed a stochastic gradient descent based TD(0) algorithm that converges for linear architectures in the *off-policy* setting. This was further extended by [7] who extended it to approximations using arbitrary smooth functions and proved convergence to a local optimum. In this section we show how to extend these off-policy methods to the robust setting with uncertain transitions. Note that this is an *alternative approach* to the requirement of Assumption 4.1, since under this assumption all off-policy methods would also converge.

The main idea of [23] is to devise stochastic gradient algorithms to minimize the following loss function called the *mean square projected Bellman error* (MSPBE) also studied in [1, 12].

$$\text{MSPBE}(\theta) := \|v_\theta - \Pi T_\pi v_\theta\|_\xi^2. \tag{95}$$

Note that the loss function is 0 for a $\theta$ that satisfies the *projected Bellman equation*, $\Phi\theta = T_\pi(\Phi\theta)$. Consider a linear architecture as in Section 4.1 where $v_\theta := \Phi\theta$. Let $i \in \mathcal{X}$ be a random state chosen with distribution $\xi_i$. Denote $\phi(i)$ by the shorthand $\phi$ and $\phi(i')$ by $\phi'$. Then it is easy to show that

$$\text{MSPBE}(\theta) := \|v_\theta - \Pi T_\pi v_\theta\|_\xi^2 = \mathbb{E}\left[d\phi\right]^\top \mathbb{E}\left[\phi\phi^\top\right]^{-1}\mathbb{E}\left[d\phi\right], \tag{96}$$

where the expectation is over the random state $i$ and $d$ is the temporal difference error for the transition $(i, i')$ i.e., $d := c(i, a) + \vartheta\theta^\top\phi' - \theta^\top\phi$, where the action $a$ and the new state $i'$ are chosen according to the exploration policy $\widehat{\pi}$. The negative gradient of the MSPBE function is

$$-\frac{1}{2}\nabla\text{MSPBE}(\theta) = \mathbb{E}\left[(\phi - \vartheta\phi')\phi^\top\right]w \tag{97}$$

$$= \mathbb{E}\left[d\phi\right] - \vartheta\mathbb{E}\left[\phi'\phi^\top\right]w \tag{98}$$

where $w = \mathbb{E}\left[\phi\phi^\top\right]^{-1}\mathbb{E}\left[d\phi\right]$. Both $d$ and $w$ depend on $\theta$. Since the expectation is hard to compute exactly [23] introduce a set of weights $w_k$ whose purpose is to estimate $w$ for a fixed $\theta$. Let $d_k$ denote the temporal difference error for a parameter $\theta_k$. The weights $w_k$ are then updated on a fast time scale as

$$w_{k+1} := w_k + \beta_k\left(d_k - \phi_k^\top w_k\right)\phi_k, \tag{99}$$

while the parameter $\theta_k$ is updated on a slower timescale in the following two possible manners

$$\theta_{k+1} := \theta_k + \alpha_k \left(\phi_k - \vartheta\phi'_k\right)\left(\phi_k^\top w_k\right) \quad \text{GTD2} \tag{100}$$

$$\theta_{k+1} := \theta_k + \alpha_k d_k \phi_k - \vartheta\alpha_k\phi'_k(\phi_k^\top w_k) \quad \text{TDC} \tag{101}$$

[7] extended this to the case of smooth nonlinear architectures, where the space $S := \left\{ v_\theta \mid \theta \in \mathbb{R}^d \right\}$ spanned by all value functions $v_\theta$ is now a differentiable sub-manifold of $\mathbb{R}^n$ rather than a linear subspace. Projecting onto such nonlinear manifolds is a computationally hard problem, and to get around this [7] project instead onto the tangent plane at $\theta$ assuming the parameter $\theta$ changes very little in one step. This allows [7] to generalize the updates of equations (99) and (100) with an additional Hessian term $\nabla^2 v_\theta$ which vanishes if $v_\theta$ is linear in $\theta$.

In the following sections we extend the stochastic gradient algorithms of [7, 24, 23] to the robust setting with uncertain transition matrices. Since the number $n$ of states is prohibitively large, we will make the simplifying assumption that $U_i^a = U$ and $\widehat{U}_i^a = U_i^a$ for the results of the following sections.

### 4.2.1 Robust stochastic gradient algorithms with linear architectures

In this section we extend the results of [23] to the robust setting, where we are interested in finding a solution to the *robust projected Bellman equation* $\Phi\theta = T_\pi\left(\Phi\theta\right)$, where $T_\pi$ is the robust Bellman operator of equation (76). Let $\widehat{T}_\pi$ denote the proxy robust Bellman operators using the proxy uncertainty set $\widehat{U}$ instead of $U$. A natural generalization of [23] is to introduce the following loss function which we call *mean squared robust projected Bellman error* (MSRPBE):

$$\text{MSRPBE}(\theta) := \left\| v_\theta - \Pi\widehat{T}_\pi v_\theta \right\|_\xi^2, \tag{102}$$

where the proxy robust Bellman operator $\widehat{T}$ is used. Note that $\widehat{T}_\pi$ is no longer truly linear in $\theta$ even for linear architectures $v_\theta = \Phi\theta$ as

$$(\widehat{T}_\pi\Phi\theta)(i) = c(i, \pi(i)) + \vartheta\sigma_{\mathcal{P}_i^{\pi(i)}}(\Phi\theta) \tag{103}$$

$$= c(i, \pi(i)) + \vartheta\theta^\top\Phi^\top p_i^{\pi(i)} + \vartheta \sup_{q \in \Phi^\top(\widehat{U})} q^\top\theta, \tag{104}$$

where $p_i^{\pi(i)}$ are the simulator transition probability vector. However, under the assumption that $\widehat{U}$ is a nicely behaved set such as a ball or an ellipsoid, so that changing $\theta$ in a small neighborhood does not lead to jumps in $\sigma_{\Phi^\top(\widehat{U})}(\theta)$, we may define the gradient $\nabla_\theta\widehat{T}_\pi(\Phi\theta)(i)$ as

$$\nabla_\theta((\widehat{T}_\pi\Phi\theta)(i)) := \vartheta\Phi^\top p_i^{\pi(i)} + \vartheta\arg\max_{q \in \Phi^\top(\widehat{U})} q^\top\theta \tag{105}$$

$$= \vartheta\arg\max_{q \in \Phi^\top\left(\widehat{\mathcal{P}_i^{\pi(i)}}\right)} q^\top\theta. \tag{106}$$

Recall the *robust temporal difference error* $\widetilde{d}$ for state $i$ with respect to the proxy set $\widehat{U}$ as in equation (47)

$$\widetilde{d} := c(i, \pi(i)) + \vartheta v_\theta(i') + \sigma_{\widehat{U}}(v_\theta) - v_\theta(i). \tag{107}$$

Under the assumption that $\mathbb{E}\left[\phi\phi^\top\right]$ is full rank, we may write the MSRPBE loss function in terms of the robust temporal difference errors $\widetilde{d}$ of equation (47) as in [23]:

$$\text{MSRPBE}(\theta) = \mathbb{E}\left[\widetilde{d}\phi\right]^\top \mathbb{E}\left[\phi\phi^\top\right]^{-1} \mathbb{E}\left[\widetilde{d}\phi\right]. \tag{108}$$

Note that if $\mathbb{E}\left[\phi\phi^\top\right]$ is full rank, then $\text{MSRPBE}(\theta) = 0$ if and only if $\mathbb{E}\left[\widetilde{d}\phi\right] = 0$ because of equation (108). Define

$$\mu_P(\theta) := \nabla\max_{y \in P} y^\top v_\theta = \nabla\max_{y \in P} y^\top\Phi\theta = \Phi^\top\arg\max_{y \in P} y^\top\theta = \arg\max_{y \in \Phi^\top(P)} y^\top\theta \tag{109}$$

for any convex compact set $P \subset \mathbb{R}^n$, so that the gradient of the MSRPBE loss function can be written as

$$-\frac{1}{2}\nabla MSRPBE(\theta) = \mathbb{E}\left[\left(\phi - \vartheta\mu_{\widehat{U}}(\theta) - \vartheta\phi'\right)\phi^\top\right]\mathbb{E}\left[\phi\phi^\top\right]^{-1}\mathbb{E}\left[\widetilde{d}\phi\right], \qquad (110)$$

$$= \mathbb{E}\left[\left(\phi - \vartheta\mu_{\widehat{U}}(\theta)\right)\phi^\top\right]w, \qquad (111)$$

$$= \mathbb{E}\left[\widetilde{d}\phi\right] - \vartheta\mathbb{E}\left[\phi'\phi^\top\right]w - \vartheta\mathbb{E}\left[\mu_{\widehat{U}}(\theta)\phi^\top\right]w \qquad (112)$$

where $w = \mathbb{E}\left[\phi\phi^\top\right]^{-1}\mathbb{E}\left[\widetilde{d}\phi\right]$ is the same as in equation (97) and [23]. Therefore, as in [23] we have an estimator $w_k$ for the weights $w$ for a fixed parameter $\theta_k$ as

$$w_{k+1} := w_k + \beta_k\left(\widetilde{d}_k - \phi_k^\top w_k\right)\phi_k, \qquad (113)$$

with the corresponding parameter $\theta_k$ being updated as

$$\theta_{k+1} := \theta_k + \alpha_k\left(\phi_k - \vartheta\mu_{\widehat{U}}(\theta) - \phi_k'\right)(\phi_k^\top w_k) \quad \text{robust-GTD2} \qquad (114)$$

$$\theta_{k+1} := \theta_k + \alpha_k\widetilde{d}_k\phi_k - \vartheta\alpha_k(\phi_k' + \mu_{\widehat{U}}(\theta))(\phi_k^\top w_k) \quad \text{robust-TDC.} \qquad (115)$$

**Run time analysis:** Let $T_n(P)$ denote the time to optimize linear functions over the convex set $P$ for some $P \subset \mathbb{R}^n$. Note that the values $v_\theta(i)$ can be computed simply in $O(d)$ time. Thus the updates of *robust-GTD2* and *robust-TDC* can be computed in $O\left(d + T_n\left(\widehat{U}\right)\right)$ time. In particular if the set $\widehat{U}$ is a simple set like an ellipsoid with associated matrix $A$, then the optimum value $\sigma_{\widehat{U}}(v_\theta)$ is simply $\sqrt{\theta^\top\Phi^\top A\Phi\theta}$, where $\Phi$ is the feature matrix. In this case we only need to compute $\Phi^\top A\Phi$ once and store it for future use. However, note that this still takes time polynomial in $n$, which is undesirable for $n \gg d$. In this case, we need to to make the assumption that there are good rank-$d$ approximations to $\widehat{U}$ i.e., $A \approx BB^\top$ for some $n \times d$ matrix $B$.

Thus the total run time for each update in this case is $O(d^2)$. If the uncertainty set is spherically symmetric, i.e., a ball, then the expression is simply $\|\Phi\theta\|_2$ and the robust temporal difference errors of equation (47) and the updates of equation (113) and (114) can be viewed simply as regular updates of [24] with an added *noise term*.

### 4.2.2 Robust stochastic gradient algorithms with nonlinear architectures

In this section we generalize the results of Section 4.2.1 where we show how to extend the algorithms of equation (113) and (114) to the case when the value function $v_\theta$ is no longer a linear function of $\theta$. This also generalizes the results of [7] to the robust setting with corresponding robust analogues of *nonlinear GTD2* and *nonlinear TDC* respectively. Let $\mathcal{M} := \left\{v_\theta \mid \theta \in \mathbb{R}^d\right\}$ be the manifold spanned by all possible value functions and let $P\mathcal{M}_\theta$ be the *tangent plane* of $\mathcal{M}$ at $\theta$. Let $T\mathcal{M}_\theta$ be the *tangent space*, i.e., the translation of $P\mathcal{M}_\theta$ to the origin. In other words, $T\mathcal{M}_\theta := \left\{\Phi_\theta u \mid u \in \mathbb{R}^d\right\}$, where $\Phi_\theta$ is an $n \times d$ matrix with entries $\Phi_\theta(i,j) := \frac{\partial}{\partial\theta_j}v_\theta(i)$. Let $\Pi_\theta$ denote the projection with to the weighted Euclidean norm $\|\cdot\|_\xi$ on to the space $T\mathcal{M}_\theta$, so that

$$\Pi_\theta = \Phi_\theta\left(\Phi_\theta\Xi\Phi_\theta\right)^{-1}\Phi_\theta^\top\Xi \qquad (116)$$

where $\Xi$ is the $n \times n$ diagonal matrix with entries $\xi_i$ for $i \in \mathcal{X}$ as in Section 4.1. The *mean squared projected Bellman equation* (MSPBE) loss function considered by [7] can then be defined as

$$\text{MSPBE}(\theta) = \|v_\theta - \Pi_\theta Tv_\theta\|_\xi^2, \qquad (117)$$

where we now project to the the tangent space $T\mathcal{M}_\theta$. The robust version of the MSPBE loss function, the *mean squared robust projected Bellman equation* (MSRPBE) loss can then be defined in terms of the *robust Bellman operator* over the proxy uncertainty set $\widehat{U}$

$$\text{MSRPBE}(\theta) = \left\|v_\theta - \Pi_\theta\widehat{T}v_\theta\right\|_\xi^2, \qquad (118)$$

and under the assumption that $\mathbb{E}\left[\nabla v_\theta(i)\nabla v_\theta(i)^\top\right]$ is non-singular, this may be expressed in terms of the *robust temporal difference* error $\widetilde{d}$ of equation (47) as in [7] and equation (108):

$$\text{MSRPBE}(\theta) = \mathbb{E}\left[\widetilde{d}\nabla v_\theta(i)\right]^\top \mathbb{E}\left[\nabla v_\theta(i)\nabla v_\theta(i)^\top\right]^{-1}\mathbb{E}\left[\widetilde{d}\nabla v_\theta(i)\right], \quad (119)$$

where the expectation is over the states $i \in \mathcal{X}$ drawn from the distribution $\xi$. Note that under the assumption that $\mathbb{E}\left[\nabla v_\theta(i)\nabla v_\theta(i)^\top\right]$ is non-singular, it follows due to equation (119) that $\text{MSRPBE}(\theta) = 0$ if and only if $\mathbb{E}\left[\widetilde{d}\nabla v_\theta(i)\right] = 0$. Since $v_\theta$ is no longer linear in $\theta$, we need to redefine the gradient $\mu$ of $\sigma$ for any convex, compact set $P$ as

$$\mu_P(\theta) := \nabla \max_{y\in P} y^\top v_\theta = \Phi_\theta^\top \arg\max_{y\in P} y^\top v_\theta, \quad (120)$$

where $\Phi_\theta(i) := \nabla v_\theta(i)$. The following lemma expresses the gradient $\nabla \text{MSRPBE}(\theta)$ in terms of the *robust temporal difference errors*, see Theorem 1 of [7] for the non-robust version.

**Lemma 4.5.** *Assume that $v_\theta(i)$ is twice differentiable with respect to $\theta$ for any $i \in \mathcal{X}$ and that $W(\theta) := \mathbb{E}\left[\nabla v_\theta(i)\nabla v_\theta(i)^\top\right]$ is non-singular in a neighborhood of $\theta$. Let $\phi := \nabla v_\theta(i)$ and define for any $u \in \mathbb{R}^d$*

$$h(\theta, u) := -\mathbb{E}\left[(\widetilde{d} - \phi^\top u)\nabla^2 v_\theta(i)u\right]. \quad (121)$$

*Then the gradient of* MSRPBE *with respect to $\theta$ can be expressed as*

$$-\frac{1}{2}\nabla\text{MSRPBE}(\theta) = \mathbb{E}\left[\left(\phi - \vartheta\mu_{\widehat{U}}(\theta) - \vartheta\phi'\right)\phi^\top\right]w + h(\theta, w), \quad (122)$$

*where $w = \mathbb{E}\left[\phi\phi^\top\right]^{-1}\mathbb{E}\left[\widetilde{d}\phi\right]$ as before.*

*Proof.* The proof is similar to Theorem 1 of [7] by using $\mu_{\widehat{U}}(\theta)$ as the gradient of $\sigma_{\widehat{U}}(\theta)$. □

Lemma 4.5 leads us to the following robust analogues of *nonlinear GTD* and *nonlinear TDC*. The update of the weight estimators $w_k$ is the same as in equation (113)

$$w_{k+1} := w_k + \beta_k\left(\widetilde{d}_k - \phi_k^\top w_k\right)\phi_k, \quad (123)$$

with the parameters $\theta_k$ being updated on a slower timescale as

$$\theta_{k+1} := \Gamma\left(\theta_k + \alpha_k\left\{\left(\phi_k - \vartheta\phi_k' - \vartheta\mu_{\widehat{U}}(\theta)\right)(\phi_k^\top w_k) - h_k\right\}\right) \qquad \text{robust-nonlinear-GTD2} \quad (124)$$

$$\theta_{k+1} := \Gamma\left(\theta_k + \alpha_k\left\{\widetilde{d}_k\phi_k - \vartheta\phi_k' - \vartheta\mu_{\widehat{U}}(\theta)(\phi_k^\top w_k) - h_k\right\}\right) \qquad \text{robust-nonlinear-TDC,} \quad (125)$$

where $h_k := \left(\widetilde{d}_k - \phi_k^\top w_k\right)\nabla^2 v_{\theta_k}(i_k)w_k$ and $\Gamma$ is a projection into an appropriately chosen compact set $C$ with a smooth boundary as in [7]. As in [7] the main aim of the projection is to prevent the parameters to diverge in the early stages of the algorithm due to the nonlinearities in the algorithm. In practice, if $C$ is large enough that it contains the set of all possible solutions $\left\{\theta\,\middle|\,\mathbb{E}\left[\widetilde{d}\nabla v_\theta(i)\right] = 0\right\}$ then it is quite likely that no projections will happen. However, we require the projection for the convergence analysis of the *robust-nonlinear-GTD2* and *robust-nonlinear-TDC* algorithms, see Section 4.2.3. Let $T_n(P)$ denote the time to optimize a linear function over the set $P \subset \mathbb{R}^n$. Then the run time is $O\left(d + T_n\left(\widehat{U}\right)\right)$. If $\widehat{U}$ is an ellipsoid with associated matrix $A$, then an approximate optimum may be computed by sampling, if we have a rank-$d$ approximation to $A$, i.e., $A \approx BB^\top$ for some $n \times d$ matrix. If $\widehat{U}$ is spherically symmetric, then the $\sigma\left(\widehat{U}\right)$ is simply $\|v_\theta\|_2$ so that the updates of equations (123) and (114) may be viewed as the regular updates of [7] with an added noise term.

### 4.2.3 Convergence analysis

In this section we provide a convergence analysis for the *robust-nonlinear-GTD2* and *robust-nonlinear-TDC* algorithms of equations (123) and (124). Note that this also proves convergence of the *robust-GTD2* and *robust-TDC* algorithms of equations (113) and (114) as a special case. Given the set $C$ let $\mathcal{C}(C)$ denote the space of all $C \to \mathbb{R}^d$ continuous functions. Define as in [7] the function $\widehat{\Gamma}: \mathcal{C}(C) \to \mathcal{C}\left(\mathbb{R}^d\right)$

$$\widehat{\Gamma}f(\theta) := \lim_{\varepsilon \to 0} \frac{\Gamma(\theta + \varepsilon f(\theta)) - \theta}{\varepsilon}. \tag{126}$$

Since $\Gamma(\theta) = \arg\min_{\theta' \in C} \|\theta - \theta'\|$ and the boundary of $C$ is smooth, it follows that $\widehat{\Gamma}$ is well defined. Let $\mathring{C}$ denote the interior of $C$ and $\partial C$ denote its boundary so that $\mathring{C} = C \setminus \partial C$. If $\theta \in \mathring{C}$, then $\widehat{\Gamma}v(\theta) = v(\theta)$, otherwise $\widehat{\Gamma}(\theta)$ is the projection of $v(\theta)$ to the tangent space of $\partial C$ at $\theta$. Consider the following ODE as in [7]:

$$\dot{\theta} = \widehat{\Gamma}\left(-\frac{1}{2}\nabla \operatorname{MSRPBE}\right)(\theta), \quad \theta(0) \in C \tag{127}$$

and let $K$ be the set of all stable equilibria of equation (127). Note that the solution set $\left\{\theta \,\middle|\, \mathbb{E}\left[\widetilde{d}\phi\right] = 0\right\} \subset K$. The following theorem shows that under the assumption of Lipschitz continuous gradients and suitable assumptions on the step lengths $\alpha_k$ and $\beta_k$ and the uncertainty set $\widehat{U}$, the updates of equations (123) and (124) converge.

**Theorem 4.6** (Convergence of *robust-nonlinear-GTD2*). *Consider the robust nonlinear updates of equations* (123) *and* (124) *with step sizes that satisfy* $\sum_{k=0}^{\infty} \alpha_k = \sum_{k=0}^{\infty} \beta_k = \infty$, $\sum_{k=0}^{\infty} \alpha_k^2, \sum_{k=0}^{\infty} \beta_k^2 < \infty$, *and* $\frac{\alpha_k}{\beta_k} \to 0$ *as* $k \to \infty$. *Assume that for every* $\theta$ *we have* $\mathbb{E}\left[\phi_\theta \phi_\theta^\top\right]$ *is non-singular. Also assume that the matrix* $\Phi_\theta$ *of gradients of the value function defined as* $\Phi_\theta(i) := \nabla v_\theta(i)$ *is Lipschitz continuous with constant* $L$, *i.e.,* $\|\Phi_\theta - \Phi_{\theta'}\|_2 \le L \|\theta - \theta'\|_2$. *Then with probability* 1, $\theta_k \to K$ *as* $k \to \infty$.

*Proof.* The argument is similar to the proof of Theorem 2 in [7]. The only thing we need to verify is the Lipschitz continuity of the robust version $\widetilde{g}(\theta_k, w_k)$ of the function $g(\theta_k, w_k)$ of [7] defined as

$$\widetilde{g}(\theta_k, w_k) := \mathbb{E}\left[(\phi_k - \vartheta\mu_{\widehat{U}}(\theta)\phi_k^\top w_k - h_k \mid \theta_k, w_k\right], \tag{128}$$

where $g(\theta_k, w_k)$ is defined as $g(\theta_k, w_k) := \mathbb{E}\left[(\phi_k - \vartheta\phi_k'(\theta)\phi_k^\top w_k - h_k \mid \theta_k, w_k\right]$, where $\phi_k'$ is the features of the state $i'$ the simulator transitions to from state $i$. Thus we only need to verify Lipschitz continuity of $\mu_{\widehat{U}}(\theta)$. Let $y^* := \arg\max_{y \in \widehat{U}} y^\top v_\theta$ and let $z^* := \arg\max_{z \in \widehat{U}} z^\top v_\theta'$.

$$\left\|\mu_{\widehat{U}}(\theta) - \mu_{\widehat{U}}(\theta')\right\|_2 = \left\|\Phi_\theta^\top y^* - \Phi_{\theta'}^\top z^*\right\|_2 \tag{129}$$

$$\le \left\|\Phi_\theta^\top y^* - \Phi_{\theta'}^\top y^*\right\|_2 \tag{130}$$

$$\le \|\Phi_\theta - \Phi_{\theta'}\|_2 \|y^*\|_2 \tag{131}$$

$$\le \|\Phi_\theta - \Phi_{\theta'}\|_2 \arg\max_{y \in \widehat{U}} \|y\|_2 \tag{132}$$

$$\le \left(L \arg\max_{y \in \widehat{U}} \|y\|_2\right) \|\theta - \theta'\|_2. \tag{133}$$

Therefore the $\mu_{\widehat{U}}(\theta)$ is Lipschitz continuous with constant $L \arg\max_{y \in \widehat{U}} \|y\|_2$. $\qquad\square$

**Corollary 4.7.** *Under the same conditions as in Theorem* 4.6, *the* robust-GTD2, robust-TDC *and* robust-nonlinear-TDC *algorithms satisfy with probability* 1 *that* $\theta_k \to K$ *as* $k \to \infty$.

Figure 2: Performance of robust models with different sizes of confidence regions on two environments. Left: **FrozenLake-v0** Right: **Acrobot-v1**

## 5    Experiments

We implemented robust versions of Q-learning, SARSA, and TD($\lambda$)-learning as described in Section 3 and evaluated their performance against the nominal algorithms using the OpenAI gym framework [10]. The environments considered for the exact dynamic programming algorithms are the text environments of **FrozenLake-v0**, **FrozenLake8x8-v0**, **Taxi-v2**, **Roulette-v0**, **NChain-v0**, as well as the control tasks of **CartPole-v0**, **CartPole-v1**, **InvertedPendulum-v1**, together with the continuous control tasks of **MuJoCo** [27]. To test the performance of the robust algorithms, we perturb the models slightly by choosing with a small probability $p$ a random state after every action. The size of the confidence region $U_i^a$ for the robust model is chosen by a 10-fold cross validation using line search. After the Q-table or the value functions are learned for the robust and the nominal algorithms, we evaluate their performance on the true environment. To compare the true algorithms we compare both the *cumulative reward* as well as the *tail distribution function* (complementary cumulative distribution function) as in [26] which for every $a$ plots the probability that the algorithm earned a reward of at least $a$.

Note that there is a tradeoff in the performance of the robust algorithms versus the nominal algorithms in terms of the value $p$. As the value of $p$ increases, we expect the robust algorithm to gain an edge over the nominal ones as long as $\widehat{U}$ is still within the simplex $\Delta_n$. Once we exceed the simplex $\Delta_n$ however, the robust algorithms decays in performance. This is due to the presence of the $\beta$ term in the convergence results, which is defined as

$$\beta := \max_{i \in \mathcal{X}, a \in \mathcal{A}} \max_{y \in \widehat{U}_i^a} \min_{x \in U_i^a} \|y - x\|_1 \,, \tag{134}$$

and it grows larger proportional to how much the proxy confidence region $\widehat{U}$ is outside $\Delta_n$. Note that while $\beta$ is 0, the robust algorithms converge to the exact Q-factor and value function, while the nominal algorithm does not. However, since large values of $\beta$ also lead to suboptimal convergence, we also expect poor performance for too large confidence regions, i.e., large values of $p$. Figure 2 depicts how the size of the confidence region affects the performance of the robust models; note that the. Note that the average score appears somewhat erratic as a function of the size of the uncertainty set, however this is due to our small sample size used in the line search. See Figures 3, 4, 5, 6, 7, 8, 9, 10, 11, and 12 for a comparison of the best robust model and the nominal model.