[Reviews · NeurIPS 2017]

Reviewer 1



The paper tackles the robust MDP setting, where the problem being solved lies within some set of possibilities, and the goal is to obtain a policy that does well in the worst case. In particular, the paper starts from the (known) robust Bellman equation and derives a number of model-free algorithms (analogs to Q-learning, SARSA, TD-learning, LSTD, GTD, and more, many with convergence guarantees in the robust MDP setting. The paper itself contains the results of a single experiment with robust Q-learning, with more in the supplemental materials. ---Quality--- This paper presents a *lot* of theoretical results -- the supplemental material is a full version of the paper and is 24 pages long! I cannot say that I have evaluated it all in full detail. That said, it does seem to me that the principle ideas that underly the new derivations are sensible and the conclusions seem reasonable. I think the empirical evaluation falls a little short. In the main paper, only one experiment with one algorithm is presented, and the results are barely discussed. I understand that space is tight, and the paper is packing a lot in. Still, I encourage the authors to try to fit a little more analysis of the experimental results in. The reader needs a little help to pull out the high points from those plots. Even in the appendix, there are more experiments, but only using Q-learning and again, there is very little in the way of interpretation of the results. All that said, I do think the results are enough to provide evidence that robust Q-learning is doing something interesting and maybe that's all that can fit in this paper. ---Clarity--- This paper is very dense, as it is essentially a survey of all of model-free reinforcement learning, but adding the robustness wrinkle at every step. Given that, I think the authors have done a good job of organizing and presenting the main ideas. I appreciated the concise intuitive descriptions of the role of each result in the broader theoretical framework. Minor points: Am I missing something or does the symbol used for the discount factor change partway through the paper? It's not a big deal, but consistency would be better. Labels on the figures are *far* too small to read. ---Originality--- As far as I am aware this paper's exploration of model-free robust RL is novel. Of course the analysis of these algorithms relies quite a bit on existing results, but I felt the paper did a good job of contextualizing its contributions. ---Significance--- I think these results are interesting and of potential practical importance. The motivating problem of training an agent on a simulator that is suspected not to be a perfect match with the environment in which the agent will be evaluated is completely plausible. In some ways I find these results more compelling than the model-based study of robust-RL, where it is often assumed that the agent knows quite a bit about the uncertainty over the model, an assumption I often find implausible. Here the agent is simply assuming that the training environment is incorrect and imposes *it's own* assumption about the uncertainty over the dynamics. Obviously its later success depends on whether that model of uncertainty was adequate. This makes a lot of sense to me!

Reviewer 2



Overview: The authors propose an extension of Robust RL methods to model-free applications. Common algorithms (SARSA, Q-learning, TD(lambda)) are extended to their robust, model-free version. The central point of the paper is replacing the true and unknown transition probabilities P_i^a with a known confidence region U_i^a centered on an unknown probability p_i^a. Transitions according to p_i^a are sampled via a simulator. The robustness of the algorithms derive then from an additional optimization step in U_i^a. The authors clarify that, since the addition of term U to p must not violate the probability simplex. However, since p_i^a is unknown, the additional optimization is actually performed on a term U_hat that ignores the above restriction. This mismatch between U and U_hat, represented by a term beta, determines whether the algorithms converge towards to an epsilon-optimal policy. The value of epsilon is also function of beta. The authors propose also an extension of RL with function approximation architectures, both linear and nonlinear. In the linear case, the authors show that a specific operator T is a contraction mapping under an opportune weighted Euclidean norm. By adopting the steady state distribution of the exploration policy as weights, and by adopting an assumption from previous literature concerning a constant alfa < 1, the authors prove that, under conditions on alfa and beta, the robust operator T_hat is a contraction, and that the solution converges close to the true solution v_pi, where the closeness depends again on alfa and beta. Furthermore, the authors extend the results of Ref.6 to the robust case for smooth nonlinear approximation architectures. An additional assumption here is to assume a constant confidence bound U_i^a = U for all state-action pairs for the sake of runtime. The authors extend on the work of Ref.6 by introducing the robust extension of the mean squared projected bellman equation, the MSRPBE, and of the GTD2 and TDC update algorithms, which converge to a local optima given assumptions on step lengths and on confidence region U_hat. In conclusion, the authors apply both nominal and robust Q-learning, SARSA and TD-learning to several OpenAI gym experiments. In these experiments, a small probability of transitioning to a random state is added to the original models during training. The results show that the best robust Q-learning outperforms nominal Q-learning. However, it is shown that the performance of the robust algorithms is not monotone with p: after increasing for small values of p, the advantage of the robust implementation decreases as U and U_hat increase in discrepancy. Evaluation: - Quality: the paper appears to be mathematically sound and assumptions are clearly stated. However, the experimental results are briefly commented. Figures are partial: of the three methods compared, Q-learning, SARSA and TD-learning, only results from Q-learning are briefly presented. The algorithms of section 4 (with function approximation) are not accompanied by experimental results, but due to the theoretical analysis preceding them this is of relative importance. - Clarity: the paper is easy to read, with the exception of a few symbolic mismatches that are easy to identify and rectify. The structure of the paper is acceptable. However, the overall clarity of the paper would benefit from extending section 4.2 and 5, possibly by reducing some portions of section 3. - Originality: the model-free extensions proposed are an original contribution with respect to existing literature and to NIPS in particular. - Significance: the motivation of the paper is clear, as is the contribution to the existing theoretical research. Less clear is the kind of application this work is intended for. In particular, the paper assumes the existence of a simulator with unknown transition probabilities, but known confidence bounds to the actual model being simulated, to obtain samples from. While this assumption is not by itself wrong, it is not clearly mentioned in the abstraction or in the introduction and takes the risk of appearing as artificial. Other comments: - Line 194: the definition of beta given in this line is the one Lemma 4.2 rather than the one of Lemma 3.2; - Line 207: here and in the following, nu is used instead of theta to indicate the discount factor. Please make sure the symbology is consistent; - Line 268: the sentence "the state are sampled with from an" might contain a typo; - Eq. 7-8: there appears to be an additional minus signs in both equations.

Reviewer 3



Summary: This paper claims to extend the theory of robust MDPs to the model-free reinforcement learning setting. However that claim ignores relevant prior work (see originality comments below). The paper also presents robust versions of Q-learning, SARSA, and TD-learning. Quality: As far as I can tell the main paper is technically sound, but I did not carefully read through the appendix. However the claim that this paper extends the theory of robust MDPs to the model-free reinforcement learning setting is incorrect. Clarity: On the one hand this paper is well written and reasonably easy to understand. On the other hand, the organization of the supplementary material is quite confusing. Rather than an appendix, the supplementary material is a longer version of the original paper. What is particularly confusing is that the numbering of lemmas and theorems, and even their statement differs between the main paper and supplementary material. Originality: The theory of robust MDPs has already been extended to the model-free reinforcement learning setting, see for example [1]. In effect, the main contribution of this paper is not original. On the other hand, as far as I know this is the first time anyone has derived robust versions of Q-learning, SARSA, and TD-learning. Significance: I expect robust versions of Q-learning, SARSA, and TD-learning to be useful for future research. It would be nice to see a comparison between the methods proposed in this paper and existing robust MDP methods for model-free reinforcement learning. Minor comments: The fonts in figure 1 are way too small to be readable, and it's very hard to differentiate between the nominal and robust graphs when printed on black and white. [1] RAAM: The Benefits of Robustness in Approximating Aggregated MDPs in Reinforcement Learning, Marek Petrik, Dharmashankar Subramanian, Conference on Neural Information Processing Systems 2014